# 🐧 PICABench: How Far Are We from Physically Realistic Image Editing?

**Yuandong Pu[1,2]** * **Le Zhuo[3,4]** **Songhao Han[5]** **Jinbo Xing[6]** **Kaiwen Zhu[1,2]** **Shuo Cao[2,7]**
**Bin Fu[2]** **Si Liu[5]** **Hongsheng Li[3]** **Yu Qiao[2]** **Wenlong Zhang[2]** **Xi Chen[8]†** **Yihao Liu[2]†**

[1]Shanghai Jiao Tong University  [2]Shanghai AI Laboratory  [3]CUHK MMLab  [4] Krea AI
[5]Beihang University  [6]Tongyi Lab  [7] USTC  [8] The University of Hong Kong

## Abstract

Image editing has achieved remarkable progress recently. Modern editing models could already follow complex instructions to manipulate the original content. However, beyond completing the editing instructions, the accompanying physical effects are the key to the generation realism. For example, removing an object should also remove its shadow, reflections, and interactions with nearby objects. Unfortunately, existing models and benchmarks mainly focus on instruction completion but overlook these physical effects. So, at this moment, *how far are we from physically realistic image editing?* To answer this, we introduce **PICABench**, which systematically evaluates physical realism across eight sub-dimension (spanning optics, mechanics, and state transitions) for most of the common editing operations (add, remove, attribute change, *etc.*). We further propose the **PICAEval**, a reliable evaluation protocol that uses VLM-as-a-judge with per-case, region-level human annotations and questions. Beyond benchmarking, we also explore effective solutions by learning physics from videos and construct a training dataset **PICA-100K**. After evaluating most of the mainstream models, we observe that physical realism remains a challenging problem with large rooms to explore. We hope that our benchmark and proposed solutions can serve as a foundation for future work moving from naive content editing toward physically consistent realism.

**Project page:** https://pica-research.github.io

## 1 Introduction

Recent advances in instruction-based image editing have brought remarkable progress (Wu et al., 2025a; Batifol et al., 2025; OpenAI, 2025b; Google, 2025; Seedream et al., 2025; Liu et al., 2025; Cai et al., 2025). In particular, with the emergence of unified multi-modal models (Deng et al., 2025; Lin et al., 2025; Wu et al., 2025b), they can seamlessly follow natural language instructions and produce visually compelling, semantically coherent edits. These systems have demonstrated strong generalization capabilities across diverse domains, establishing a new standard for controllable and high-quality image manipulation.

However, the realism of image editing depends not only on semantic accuracy but also on the correct rendering of physical effects. Even simple operations like object addition or removal often trigger complex interactions with lighting, shadows, and object support in the scene. Existing benchmarks overlook this limitation by solely emphasizing semantic fidelity and visual consistency. Although some recent benchmarks (Wu et al., 2025c; Li et al., 2025) attempt to probe scientific-plausible editing capabilities, their test cases diverge from common user-edit scenarios but focus on scientific domains with specific physical or chemistry knowledge. Consequently, we lack a clear understanding of *how far we are from physically realistic image editing.*

---

*This work was done during his internship at Shanghai Artificial Intelligence Laboratory.
†Corresponding Authors

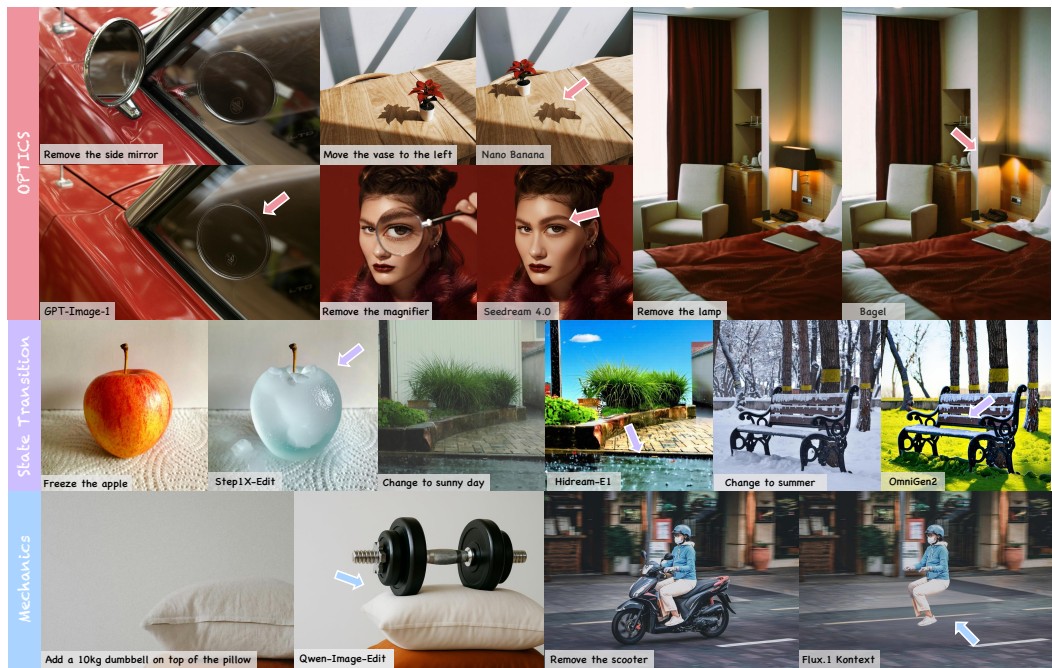

Figure 1: **Challenging cases from PICABench.** Despite providing instruction-aligned outputs, current SoTA models still struggle with generating physically realistic edits, resulting in unharmonized lighting, deformation, or state transitions with *common editing operations*.

To address this gap, we introduce **PICA** (**P**hys**IC**s-**A**ware) Bench—a diagnostic benchmark designed to evaluate physical realism in image editing beyond semantic fidelity. Drawing on common requirements in real-world editing applications Taesiri et al. (2025), we categorize physical consistency into three intuitive dimensions that are often overlooked in typical editing tasks: *Optics*, *Mechanics*, and *State Transition*. These dimensions were selected to reflect common but under-penalized error types, such as unrealistic lighting effects, impossible object deformations, or implausible state changes. Together, they span eight sub-dimensions, each defined by concrete, checkable criteria: *Optics* includes light propagation, reflection, refraction, and light-source effects; *Mechanics* captures deformation and causality; and *State Transition* addresses both global and local state changes. This fine-grained taxonomy facilitates systematic assessment of whether edited images adhere to principles such as lighting consistency, structural plausibility, and realistic state transitions. Together, it enables comprehensive evaluation and targeted diagnosis of physics violations in image editing models.

With the carefully curated test cases, evaluating the physical correctness remains challenging. We introduce **PICAEval**, a reliable and interpretable protocol tailored for physics-aware assessment. While existing VLM-as-Judge setups (Wu et al., 2025c; Niu et al., 2025; Sun et al., 2025; Zhao et al., 2025) offer a convenient way to automate evaluation, they typically rely on general prompts without grounding in physical principles. As a result, these setups often lack sensitivity to nuanced physical violations and may produce hallucinated judgments when faced with subtle or localized cues. Facing this challenge, PICAEval adopts targeted, per-example **Q&A** aligned with specific physical sub-dimensions, substantially improving diagnostic accuracy. To further reduce hallucination, we incorporate **grounded human-annotated key regions** (e.g., reflection surfaces, contact interfaces), directing the model's attention to physically relevant evidence. This protocol yields high agreement with human assessments, offering a reliable measurement for physical correctness.

Beyond evaluation, we provide a strong baseline by learning physics from videos. Specifically, we present **PICA-100K**, a synthetic dataset of 100k editing examples constructed from videos. Prior work (Yu et al., 2025b; Chen et al., 2025; Chang et al., 2025; Cao et al., 2025a) has shown that editing pairs derived from videos can enhance the quality and robustness of editing models. Motivated by recent advances in video generation approaching world-simulator (Wan et al., 2025), we design an automatic pipeline that integrates a text-to-image model as a scene renderer and an image-to-video model as a state-transition simulator. From the generated videos, we extract temporally coherent editing pairs and further recalibrate multi-level editing instructions using GPT-5. Our experiments

shows that finetuning on PICA-100K significantly improves the baseline model's capability to generate physically realistic editing results without sacrificing semantic quality.

We benchmark 11 open- and closed-source image editing models across diverse architectures and scales. PICABench comprehensively distinguishes models based on their level of physical awareness, while PICA-100K effectively improves model performance. As shown in Fig. 1, modeling physical realistic transformations is still challenging for current SoTA models, which underlines the significance of advancing from semantic editing toward physically grounded image manipulation in the future. Our main contributions could be summarized as follows.

- We introduce **PICABench**, a comprehensive and fine-grained benchmark for physics-aware image editing. It covers diversified physical effects (eight sub-dimensions) and includes the great majority of commonly required editing operations in practical applications.
- We propose **PICAEval**, a region-aware, VQA-based evaluation protocol that incorporates human-annotated key regions to provide interpretable and reliable assessments for physical correctness, improving robustness to subtle errors compared to general scoring prompts.
- We construct **PICA-100K**, a large-scale dataset derived from synthetic videos, and show that fine-tuning existing models (*e.g.* FLUX.1 Kontext) on this dataset effectively enhances their physical consistency while preserving semantic fidelity.

## 2 RELATED WORK

### 2.1 INSTRUCTION-BASED IMAGE EDITING MODELS

Recent advances in instruction-based image editing have led to substantial progress in controllable and diverse visual manipulation (Ye et al., 2025a; Yu et al., 2025a; Zeng et al., 2025; Jin et al., 2024; Huang et al., 2024; Krojer et al., 2024). Prior approaches implement image editing in a training-free manner (Yang et al., 2023; Pan et al., 2023; Couairon et al., 2022). Recent training-based methods such as HiDream-E1.1 (Cai et al., 2025), Step1X-Edit (Liu et al., 2025), FLUX.1 Kontext (Batifol et al., 2025), and Qwen-Image-Edit (Wu et al., 2025a) improve edit quality, responsiveness, and instruction alignment, while unified frameworks (e.g., Bagel (Deng et al., 2025), OmniGen2 (Wu et al., 2025b), UniWorld-V1 (Lin et al., 2025)) integrate instruction-following, visual reasoning, and multi-task learning to support diverse tasks like free-form manipulation, future-frame prediction, multiview synthesis, segmentation, and composition. Closed-source systems (e.g., GPT-Image-1 (OpenAI, 2025b), Seedream 4.0 (Seedream et al., 2025), Nano-Banana (Google, 2025)) further demonstrate strong user-intent alignment and high visual fidelity across text-to-image and image-to-image workflows. However, despite these gains, most approaches prioritize semantic and perceptual quality and often neglect physical constraints, leading to artifacts such as unrealistic shadows, refractions, and deformations, underscoring the need for physics-aware editing.

### 2.2 INSTRUCTION-BASED IMAGE EDITING BENCHMARKS

Instruction-based image editing benchmarks have evolved from early reliance on semantic (DINO, CLIP (Zhang et al., 2023; Wang et al., 2023; Ma et al., 2024)) and pixel-level metrics, which capture similarity but miss fine-grained semantic alignment, to modern "VLM-as-a-Judge" evaluations (Wu et al., 2025c; Niu et al., 2025; Zhao et al., 2025; Sun et al., 2025; Ye et al., 2025b; Liu et al., 2025; Cao et al., 2025b; Zhuo et al., 2025) that use vision-language models to rate instruction adherence, perceptual quality, and realism across diverse, complex prompts. While these LLM-based approaches enable general multi-dimensional scoring, they are prone to overlooking physically implausible edits (e.g., unrealistic lighting, deformations, or object interactions) and can hallucinate, allowing visually appealing yet inconsistent outputs to score well. To close this gap, we introduce a physics-aware benchmark and the PICAEval—a region-grounded, QA-based metric that evaluates physical consistency through localized, interpretable assessments anchored to specific regions of interest.

## 3 METHOD

In this section, we first give an overall introduction of PICABench, a benchmark structured to evaluate physical realism in image editing. We then dive into the construction steps, begin with

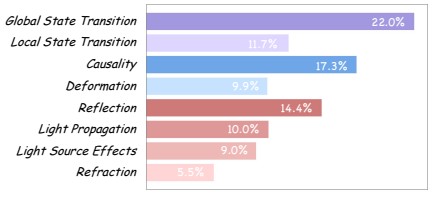 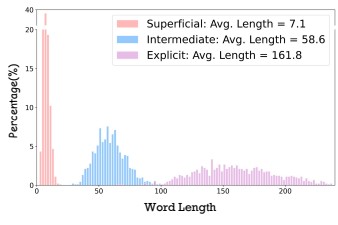 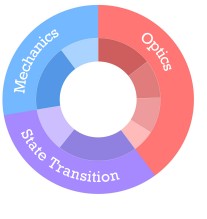

(a) Distribution of qa pairs     (b) Word length distribution of edit instruction     (c) Distribution of PICABench

Figure 2: **Statistics Analysis of PICABench.** PICABench is a comprehensive benchmark designed to evaluate physical realism of image editing models across eight sub-dimentinons. Fig. 2(a) shows distribution of QA pairs. Fig. 2(b) presents words length distribution of editing instruction across three levels of prompt. Fig. 2(c) provides a perspective on overall composition of PICABench.

the data curation pipeline, which pairs diverse images with multi-level editing instructions. Next, we present PICAEval, a region-grounded evaluation protocol for reliable assessment. Finally, we propose PICA-100K, a synthetic dataset built from videos, and show how fine-tuning on it provides a strong baseline for improving physics-aware editing.

## 3.1 PICABENCH

We introduce the task coverage and overall statistics of PICABench. Our benchmark focuses on three core dimensions of physical realism: *Optics*, *Mechanics*, and *State Transition*, which reflect common yet overlooked failure modes such as unrealistic lighting, implausible deformations, and invalid state changes. As shown in Fig. 2(c) and Fig. 3, the benchmark includes 900 editing samples spanning these three dimensions, further divided into eight sub-dimensions with concrete and checkable criteria—ranging from optical effects, to mechanical plausibility, and to realistic state transitions.

**Optics.** This category evaluates whether edited images follow the basic physical rules of light, including how it casts shadows, reflects from surfaces, bends through transparent materials, and interacts with light sources. Edits should produce shadows, reflections, refractions, and light-source effects that align with the scene's geometry and lighting—matching shadow direction and occlusion, enabling view- and shape-dependent reflections, ensuring smooth background distortion through transparent media, and maintaining consistent color, softness, and falloff for added light sources. These effects, while often subtle, are key to making edits appear natural and physically believable.

**Mechanics.** This category evaluates whether edited objects remain mechanically and causally consistent with the scene. Deformation should follow material properties—rigid objects must retain shape, while elastic ones deform smoothly with consistent texture and geometry. Causality covers a broader range of physically plausible effects, including structural responses to force redistribution, agent reactions to added or removed stimuli, and environmental changes that alter object behavior, all of which must follow consistent physical or behavioral laws.

**State transition.** This category evaluates whether environmental and material changes unfold in a physically coherent manner, either across the entire scene or within localized regions. Global state transitions, such as changes in time of day, season, or weather, must update all relevant visual cues consistently—ranging from lighting and shadows to vegetation, surface conditions, and atmospheric effects. These changes require coordinated, scene-wide modifications that follow natural temporal or environmental progression. Local state transitions, on the other hand, involve targeted physical changes confined to specific objects or regions. These include phenomena such as wetting, drying, melting, burning, freezing, wrinkling, splashing, or fracturing. Edits must integrate smoothly with surrounding context, preserve material boundaries, and maintain plausible causal triggers.

## 3.2 DATA CURATION

To enable reliable, fine-grained evaluation of physically realistic image editing, we curate benchmark entries that pair natural images with editing instructions explicitly designed to test physical consistency. Our data curation pipeline is aligned with the taxonomy in Sec. 3.1 and structured into two stages: *Data Collection* and *Edit Instruction Construction*. A visual overview is shown in Fig. 4.

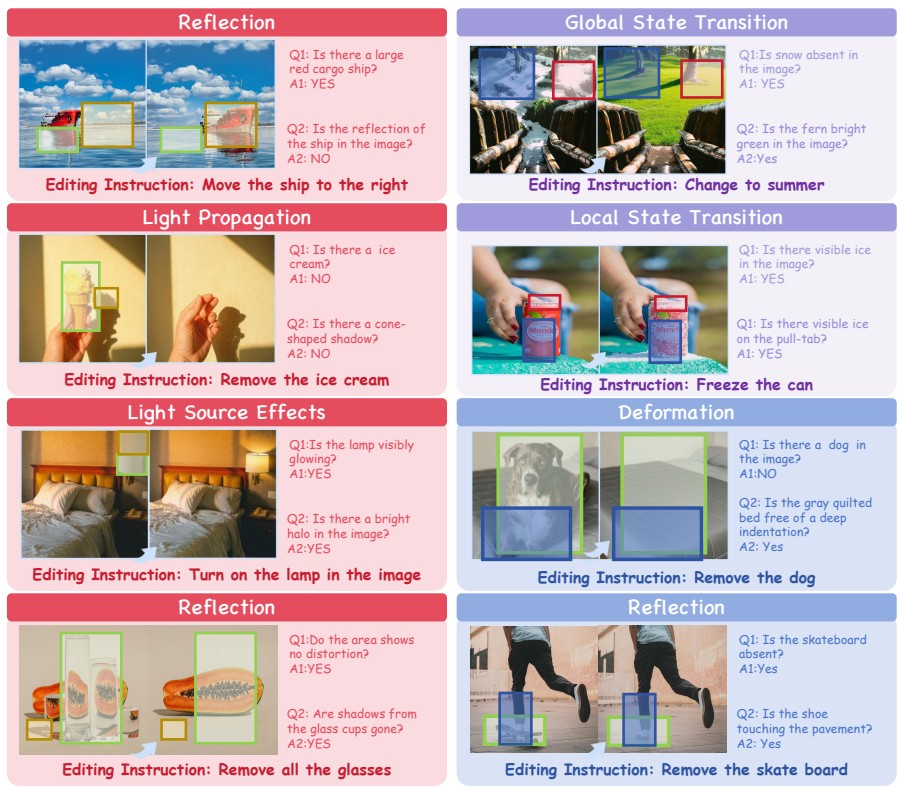

Figure 3: **Overview of PICABench.** We present illustrative examples from eight sub-dimensions. Key regions are annotated to help reduce hallucination for VLMs.

**Data collection.** We begin by defining a structured vocabulary mapped to the eight sub-dimensions. To broaden the coverage, we use GPT-5 to expand this vocabulary into a rich keyword set encompassing materials, lighting contexts, and long-tail phenomena. We then use these keywords to retrieve candidate images from licensed and public sources. We prioritize visually diverse scenes that exhibit salient physical cues, such as directional lighting, transparent or reflective media, deformable objects, or phase-changeable substances. Human annotators filter duplicates and artifacts and tag applicable sub-dimensions for each image to support subsequent annotation.

**Instruction construction.** Each retained image is paired with a human-written natural language instruction that induces a physics-relevant edit, grounded in the scene's physical affordances and designed to implicitly target a specific sub-dimension. To assess not only whether models can follow surface-level commands but also whether they can internalize and apply physical knowledge under varying prompt conditions, we construct three levels of instruction complexity: *superficial* prompts that issue plain edit commands without explanations, which probe models' intrinsic physical priors and align with realistic usage scenarios; *intermediate* prompts that include a brief rationale grounded in physical rules, blueserving as reasoning cues to activate physical knowledge; and *explicit* prompts that further describe the expected results of the edit, minimizing ambiguity to strictly assess visua capabilities. We use GPT-5 to expand each human-authored instruction into these three forms, followed by manual review to ensure clarity, factual correctness, and alignment with the visual context. For each sample, the benchmark retains a canonical version of the instruction.

### 3.3 PICAEVAL

Evaluating physically realistic image editing remains challenging. Unlike semantic fidelity or perceptual quality, physical realism is inherently contextual: it depends not only on the edited content but also on its alignment with the physical constraints implied by the original scene and instruction. Moreover, there is no reference image to serve as ground truth, and general prompting strategies such as "Is this edit correct?" often yield vague or hallucinated responses from VLMs.

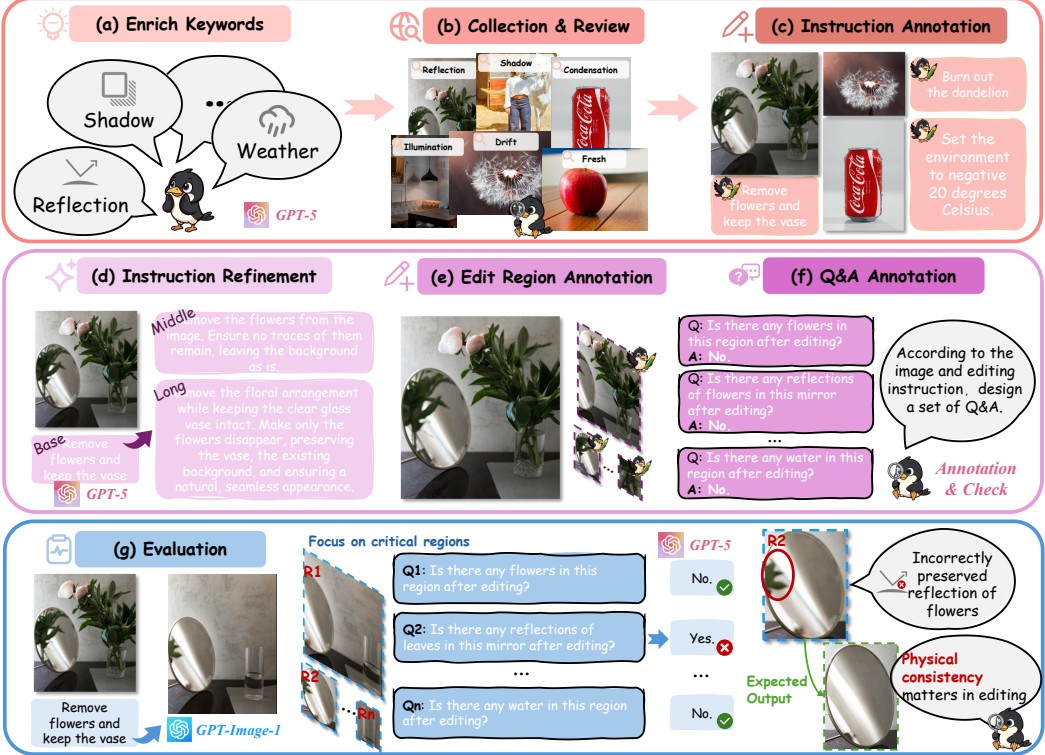

Figure 4: **Overall pipeline for benchmarks construction and evaluation**. (a–b) We enrich a physics-specific keyword set and retrieve diverse candidate images. (c–d) Human-written editing instructions are expanded into three levels of complexity using GPT-5. (e) Annotators mark physics-critical regions. (f) Spatially grounded yes/no questions are generated to evaluate physical plausibility. (g) During evaluation, VLMs answer each question with reference to the edited region.

To address this, we introduce **PICAEval**, a region-grounded, question-answering based metric designed to assess physical realism in a modular, interpretable manner. Inspired by recent work (Chai et al., 2024; Han et al., 2025), PICAEval decomposes each evaluation instance into multiple region-specific verification questions that can be reliably judged by a VLM. Each benchmark entry is paired with a curated set of spatially grounded yes/no questions designed to probe whether the edited image preserves physical plausibility within key regions. These questions are tied to observable physical phenomena—such as shadows, reflections, object contact, or material deformation—and are anchored to human-annotated regions of interest (ROIs). This design encourages localized, evidence-based reasoning and reduces the influence of irrelevant image content on the VLM's judgment.

**Evaluation pipeline.** As illustrated in Fig. 4(e–f), the evaluation proceeds as follows: (1) Annotators mark key regions in the input image where physics-critical evidence is expected to appear post-editing (e.g., reflective surfaces, deformation zones, cast shadows); (2) Using the edit instruction and region, GPT-5 generates a set of 4–5 binary QA pairs per entry, which are then manually reviewed for clarity and coverage; (3) At test time, a VLM (e.g., GPT-5) is prompted with the edited image, instruction, region, and question, and produces an answer constrained to the visible content within the region.

PICAEval is computed as the proportion of questions for which the VLM answer exactly matches the reference label. Compared to direct prompting, this QA-based protocol offers three key advantages: (i) spatial grounding reduces hallucination, (ii) decomposition increases interpretability and robustness, and (iii) the format better mirrors how humans evaluate physical plausibility—through concrete, localized checks. We report quantitative comparisons and per-subdimension breakdowns to enable diagnostic analysis of physics-aware image editing capabilities in Sec. 4.

## 3.4  STRONG BASELINE: LEARNING PHYSICAL REALISM FROM VIDEOS

To address the limitations identified in Sec. 3.1, we introduce **PICA-100K**, a purely synthetic dataset designed to improve physics-aware image editing. Our decision to use fully generated data is driven

Figure 5: **PICA-100K construction pipeline**. We first curate structured prompts for scene and subject composition, refined by GPT-5 and rendered using FLUX.1-Krea-dev for text-to-image generation. Motion-based edit instructions are created via GPT-5 and applied using Wan2.2-14B to synthesize short videos depicting physical transformations.

by three primary motivations. **First**, prior work (Yu et al., 2025b; Chen et al., 2025; Cao et al., 2025a; Chang et al., 2025) has demonstrated that constructing image-editing data from video is an effective strategy for enhancing model performance, particularly for capturing real world dynamics. **Second**, building large-scale, real-world datasets tailored to physics-aware editing is prohibitively expensive and labor-intensive. **Third**, the rapid progress in generative modeling has unlocked new possibilities: state-of-the-art text-to-image models (Labs, 2024) can now generate highly realistic and diverse images, while powerful image-to-video (I2V) models such as Wan2.2-14B (Wan et al., 2025) simulate complex dynamic processes with remarkable physical fidelity. Together, these generative priors enable the creation of training data with precise and controllable supervision signals, which are essential for training models to perform fine-grained, physically realistic edits. We find that fine-tuning the baseline on PICA-100K enhances the model's performance in real-world evaluation.

**PICA-100K dataset.** As shown in Fig. 5, we begin by constructing two structured prompt dictionaries: a Subject Dictionary and a Scene Dictionary, which include a wide array of subjects and environments (e.g., "a tea pot," "a black kitchen table"). These entries are paired using handcrafted text-to-image (T2I) templates and further refined using GPT-5, resulting in high-quality natural language instructions. The refined instructions are passed to the FLUX.1-Krea-dev (Lee et al., 2025) to generate static source images that are both visually realistic and semantically diverse.

Next, we generate motion-oriented instructions to simulate physical edits. This is accomplished by designing a series of I2V instruction templates, describing plausible motion-based changes such as rotations, movements, or tilts. These templates are expanded using GPT-5 to improve clarity and behavioral precision. The motion instructions (e.g., "remove the tea pot," "tilt the vase until it tips over," or "swing the lantern gently in the wind") are then applied to the corresponding images using Wan2.2-14B-I2V, which synthesizes short video clips depicting the intended physical transformations.

For each video, we extract the first and last frames to construct a (source, edited) image pair. These pairs, along with the corresponding instruction, are used to form supervision signals. GPT-5 is employed to annotate each pair automatically, labeling the final frame as the preferred output. This pipeline eliminates the need for manual labeling while maintaining high annotation consistency.

Our final dataset contains 105,085 instruction-based editing samples distributed across eight physics categories. The experimental results in Sec. 4 demonstrate that this pipeline can effectively generate high-quality data, significantly enhancing model performance on physics-aware image editing tasks.

**Comparison with related works.** PICA-100K is closely related to recent efforts Chang et al. (2025); Rotstein et al. (2025) that utilize video priors on image editing tasks. It differs from them in both motivation and methodology. ByteMorph (Chang et al., 2025) is primarily designed for non-rigid image editing, emphasizing visually salient motions such as articulation, deformation, and large pose or viewpoint changes. However, focus on large motions may hurt models' ability to keep non-edited region unchanged. Rotstein et al. (2025) proposes a training-free method, which focuses on zero-shot feasibility. It directly leverages a video generation model to simulate the editing process. Our work instead targets physical realism, which represents implicit physics principle of real world. Also, our data pipeline allows for more controllable generation where the non-edited regions remain stable.

**Training paradigm.** To demonstrate the effectiveness of PICA-100K, we fine-tune two representative instruction-based image editing backbones: FLUX.1-Kontext-dev (Batifol et al., 2025) and Qwen-Image-Edit (Wu et al., 2025a). We employ LoRA (Hu et al., 2022) with a rank of 256 for fine-tuning.

Table 1: **Quantitative comparison on PICABench-Superficial** evaluated by GPT-5 for instruction-based editing models, where Acc ↑, Con ↑, LP, LSE, GST, LST denote Accuracy (%) and Consistency (dB), Light propagation, Light Source Effects, Global State Transition, Local State Transition respectively. ▉ and ▉ indicates the best and second best score in a category, respectively.

| Model | LP | | LSE | | Reflection | | Refraction | | Deformation | | Causality | | GST | | LST | | Overall | |
|---|---|---|---|---|---|---|---|---|---|---|---|---|---|---|---|---|---|---|
| | Acc ↑ | Con ↑ | Acc ↑ | Con ↑ | Acc ↑ | Con ↑ | Acc ↑ | Con ↑ | Acc ↑ | Con ↑ | Acc ↑ | Con ↑ | Acc ↑ | Con ↑ | Acc ↑ | Con ↑ | Acc ↑ | Con ↑ |
| Nano Banana | 60.29 | 27.37 | 59.30 | 28.14 | 66.94 | 25.52 | 53.95 | 25.36 | 59.90 | 24.81 | 55.27 | 25.96 | 60.60 | 13.55 | 59.88 | 24.70 | 59.87 | 23.47 |
| GPT-Image-1 | 61.26 | 16.82 | 66.04 | 15.71 | 62.39 | 17.20 | 59.21 | 17.00 | 59.66 | 17.62 | 52.88 | 17.21 | 70.75 | 10.62 | 59.04 | 15.01 | 61.08 | 15.48 |
| Seedream 4.0 | 62.71 | 25.28 | 65.50 | 27.47 | 65.77 | 24.86 | 53.51 | 26.55 | 59.17 | 24.60 | 53.45 | 26.66 | 65.12 | 11.17 | 66.11 | 28.54 | 61.91 | 23.26 |
| Nano Banana Pro | 60.53 | 27.40 | 70.62 | 23.78 | 70.32 | 27.02 | 57.02 | 27.46 | 64.79 | 25.55 | 58.65 | 26.44 | 72.74 | 11.30 | 70.27 | 23.30 | 66.16 | 22.97 |
| GPT-Image-1.5 | 62.95 | 23.54 | 73.15 | 22.21 | 68.13 | 24.56 | 62.28 | 23.89 | 65.53 | 25.24 | 58.51 | 26.16 | 74.39 | 11.34 | 71.52 | 22.97 | 67.05 | 21.73 |
| DiMOO | 46.00 | 24.08 | 29.38 | 26.68 | 43.68 | 22.12 | 35.53 | 20.76 | 39.36 | 25.53 | 36.71 | 23.19 | 22.52 | 22.56 | 40.54 | 25.44 | 35.66 | 23.70 |
| Uniworld-V1 | 42.37 | 18.50 | 34.50 | 19.96 | 46.04 | 18.59 | 46.05 | 17.48 | 40.10 | 18.82 | 39.52 | 18.11 | 22.85 | 17.62 | 39.50 | 19.41 | 37.68 | 18.48 |
| Bagel | 46.97 | 34.12 | 39.35 | 35.53 | 49.41 | 33.11 | 42.54 | 28.36 | 44.25 | 33.12 | 39.24 | 33.51 | 46.80 | 10.48 | 49.27 | 30.53 | 45.07 | 28.42 |
| Bagel-Think | 49.88 | 32.44 | 50.40 | 29.10 | 47.05 | 33.37 | 43.42 | 28.87 | 49.88 | 27.59 | 38.68 | 32.88 | 45.70 | 11.66 | 50.94 | 27.28 | 46.48 | 26.88 |
| OmniGen2 | 49.64 | 25.69 | 48.79 | 28.34 | 56.49 | 27.78 | 39.04 | 24.84 | 44.74 | 29.28 | 39.80 | 26.93 | 51.10 | 12.18 | 39.09 | 25.89 | 46.79 | 24.12 |
| Hidream-E1.1 | 49.15 | 22.38 | 48.25 | 22.87 | 49.07 | 20.44 | 46.49 | 22.68 | 44.50 | 21.16 | 40.51 | 21.36 | 56.40 | 9.20 | 40.33 | 19.66 | 47.90 | 18.91 |
| Step1X-Edit | 45.04 | 30.38 | 47.44 | 27.53 | 53.46 | 29.32 | 34.21 | 32.37 | 45.72 | 29.71 | 42.90 | 30.92 | 55.85 | 8.75 | 46.57 | 20.92 | 48.23 | 24.68 |
| Flux.1 Kontext | 54.96 | 27.58 | 57.41 | 25.97 | 57.50 | 26.92 | 36.40 | 26.76 | 51.83 | 28.86 | 38.12 | 29.69 | 48.79 | 12.52 | 47.61 | 25.70 | 48.93 | 24.57 |
| Flux.1 Kontext+$SFT$ | 57.38 | 27.90 | 58.49 | 26.58 | 63.07 | 26.99 | 36.40 | 27.01 | 53.30 | 29.19 | 41.07 | 29.54 | 47.02 | 14.44 | 49.27 | 27.01 | 50.64 | 25.23 |
| Δ Improvement | +2.42 | +0.32 | +1.08 | +0.61 | +5.57 | +0.07 | +0.00 | +0.25 | +1.47 | +0.33 | +2.95 | -0.15 | -1.77 | +1.92 | +1.66 | +1.31 | +1.71 | +0.66 |
| Qwen-Image-Edit | 62.95 | 19.87 | 61.19 | 23.07 | 62.90 | 21.56 | 55.26 | 23.72 | 48.66 | 21.49 | 48.95 | 22.65 | 67.33 | 10.19 | 54.89 | 20.26 | 58.29 | 19.43 |
| Qwen-Image-Edit+$SFT$ | 61.30 | 23.24 | 68.26 | 21.91 | 71.20 | 25.33 | 56.01 | 29.73 | 57.51 | 23.66 | 54.19 | 25.51 | 65.54 | 10.57 | 59.66 | 25.60 | 62.13 | 21.91 |
| Δ Improvement | -1.65 | +3.37 | +7.07 | -1.16 | +8.30 | +3.77 | +0.75 | +6.01 | +8.85 | +2.17 | +5.24 | +2.86 | -1.79 | +0.38 | +4.77 | +5.34 | +3.84 | +2.48 |

Table 2: **Performance across different prompt specificity levels.** Model performance improves with prompt specificity, indicating that more detailed prompts yield higher performance.

| Model | LP | | LSE | | Reflection | | Refraction | | Deformation | | Causality | | GST | | LST | | Overall | |
|---|---|---|---|---|---|---|---|---|---|---|---|---|---|---|---|---|---|---|
| | Acc ↑ | Con ↑ | Acc ↑ | Con ↑ | Acc ↑ | Con ↑ | Acc ↑ | Con ↑ | Acc ↑ | Con ↑ | Acc ↑ | Con ↑ | Acc ↑ | Con ↑ | Acc ↑ | Con ↑ | Acc ↑ | Con ↑ |
| Bagel-superficial | 46.97 | 34.12 | 39.35 | 35.53 | 49.41 | 33.11 | 42.54 | 28.36 | 44.25 | 33.12 | 39.24 | 33.51 | 46.80 | 10.48 | 49.27 | 30.53 | 45.07 | 28.42 |
| Bagel-intermediate | 55.93 | 23.78 | 61.73 | 18.94 | 57.50 | 28.91 | 47.37 | 21.08 | 49.88 | 22.60 | 44.87 | 26.09 | 57.51 | 8.81 | 56.13 | 23.15 | 54.06 | 21.14 |
| Bagel-explicit | 62.71 | 15.39 | 72.24 | 13.89 | 62.39 | 18.74 | 55.26 | 16.68 | 57.70 | 15.24 | 59.35 | 19.98 | 77.26 | 8.14 | 65.90 | 15.65 | 65.61 | 15.20 |
| Flux.1 Kontext-superficial | 54.96 | 27.58 | 57.41 | 25.97 | 57.50 | 26.92 | 36.40 | 26.76 | 51.83 | 28.86 | 38.12 | 29.69 | 48.79 | 12.52 | 47.61 | 25.70 | 48.93 | 24.57 |
| Flux.1 Kontext-intermediate | 58.60 | 25.61 | 63.61 | 23.91 | 61.21 | 26.70 | 33.33 | 26.81 | 54.03 | 26.58 | 49.23 | 26.70 | 53.42 | 12.81 | 49.69 | 25.61 | 53.77 | 23.42 |
| Flux.1 Kontext-explicit | 61.74 | 25.61 | 68.19 | 21.20 | 63.24 | 25.89 | 42.98 | 24.89 | 58.44 | 24.75 | 62.59 | 23.54 | 70.75 | 10.70 | 61.75 | 23.68 | 63.30 | 21.54 |
| Qwen-Image-Edit-superficial | 62.95 | 19.87 | 61.19 | 23.07 | 62.90 | 21.56 | 55.26 | 23.72 | 48.66 | 21.49 | 48.95 | 22.65 | 67.33 | 10.19 | 54.89 | 20.26 | 58.29 | 19.43 |
| Qwen-Image-Edit-intermediate | 62.47 | 20.37 | 65.23 | 20.28 | 66.61 | 22.49 | 44.74 | 25.54 | 54.77 | 24.26 | 50.49 | 22.36 | 67.44 | 10.91 | 60.91 | 22.78 | 60.41 | 20.14 |
| Qwen-Image-Edit-explicit | 65.62 | 17.78 | 71.43 | 18.81 | 64.59 | 20.45 | 51.75 | 24.29 | 58.44 | 21.14 | 66.39 | 19.90 | 78.81 | 9.76 | 69.65 | 19.33 | 68.02 | 17.96 |

Both models are trained with an effective global batch size of 64 and optimized using AdamW for 10,000 optimization steps on 16 NVIDIA A100 GPUs. The learning rate is set to $10^{-5}$ for FLUX.1-Kontext-dev and $10^{-4}$ for Qwen-Image-Edit.

# 4 EXPERIMENT

## 4.1 EVALUATION DETAILS

We evaluate 13 closed- and open-source models, covering most recent image-editing and unified vision-language systems. Closed-source systems include GPT-Image-1 (OpenAI, 2025b), GPT-Image-1.5 (OpenAI, 2025c), Nano Banana, Nano Banana Pro (Google, 2025), and Seedream 4.0 (Seedream et al., 2025). Open-source baselines include FLUX.1 Kontext (Batifol et al., 2025), Step1X-Edit (Liu et al., 2025), Bagel (Deng et al., 2025), Bagel-Think (Deng et al., 2025), HiDream-E1.1 (Cai et al., 2025), UniWorld-V1 (Lin et al., 2025), OmniGen2 (Wu et al., 2025b), Qwen-Image-Edit (Wu et al., 2025a), and DiMOO (Xin et al., 2025). All input images are resized proportionally to a maximum resolution of 1024 on the longer side prior to evaluation. To ensure fairness and reproducibility, we run all models using their default settings from official repositories or web APIs. We choose superficial prompts as our default setting.

For PICAEval, we first use the provided annotation masks to crop the edited region from the image. The cropped region is then resized proportionally to 1024 on the longer side before being passed to the VQA-based evaluator. This ensures standardized input size while preserving relevant physical cues within the editing region. We report results using both the current state-of-the-art closed-source model (GPT-5 (OpenAI, 2025a)) and the leading open-source alternative (Qwen2.5-VL–72B (Bai et al., 2025)) as VLM evaluator. For consistency evaluation, we compute PSNR over the non-edited

Table 3: **Ablation Results.** We construct a real-video-based dataset (Mira400K). The model trained on Mira400K underperforms, highlighting the effectiveness of our targeted synthetic data pipeline.

| Model | LP | | LSE | | Reflection | | Refraction | | Deformation | | Causality | | GST | | LST | | Overall | |
|---|---|---|---|---|---|---|---|---|---|---|---|---|---|---|---|---|---|---|
| | Acc ↑ | Con ↑ | Acc ↑ | Con ↑ | Acc ↑ | Con ↑ | Acc ↑ | Con ↑ | Acc ↑ | Con ↑ | Acc ↑ | Con ↑ | Acc ↑ | Con ↑ | Acc ↑ | Con ↑ | Acc ↑ | Con ↑ |
| Flux.1 Kontext | 54.96 | 27.58 | 57.41 | 25.97 | 57.50 | 26.92 | 36.40 | 26.76 | 51.83 | 28.86 | 38.12 | 29.69 | 48.79 | 12.52 | 47.61 | 25.70 | 48.93 | 24.57 |
| +PICA100K | 57.38 | 27.90 | 58.49 | 26.58 | 63.07 | 26.99 | 36.40 | 27.01 | 53.30 | 29.19 | 41.07 | 29.54 | 47.02 | 14.44 | 49.27 | 27.01 | 50.64 | 25.23 |
| +MIRA400K | 52.06 | 28.80 | 54.72 | 29.92 | 57.34 | 27.93 | 37.28 | 28.05 | 49.88 | 29.48 | 38.68 | 32.41 | 44.59 | 40.17 | 41.58 | 31.22 | 46.96 | 32.08 |

regions by masking out the predicted edit area, thereby measuring how well models preserve the original content outside the editing scope. See Appendix A.3 for details.

## 4.2 BENCHMARK RESULTS

**We are still far from physically realistic image editing.** Tab. 1 presents a comprehensive evaluation of existing methods. While several closed-source models surpass 60 overall accuracy, the performance remains far from saturated, leaving substantial headroom for physical realistic image editing. Meanwhile, all open-source models still score below 60, indicating a persistent gap in the ability of current image editing models to generate physically realistic outputs.

**The gap between understanding and physical realism.** Among open-source models, unified architectures consistently underperform compared to dedicated image editing models. Although unified MLLMs attempt to integrate visual understanding and generation within a single framework, the presumed advantage of enhanced world understanding does not translate into improved physical realism. This suggests that stronger understanding alone is insufficient, and effectively coupling understanding with generation remains an open challenge. Tab. 2 presents performance across different prompt specificity levels. As shown in Tab. 2, model accuracy improves as prompts become more detailed. The decrease of consistency can be attributed to the trade-off between improving physical realism and preserving non-edited image regions. However, the gain from intermediate prompts is smaller than that from explicit prompts. We speculate this is due to the lack of internalized physics principles, which prevents models from leveraging the additional information. Interestingly, the Bagel model outperforms Flux Kontext under explicit prompts, likely because its unified architecture enhances long-text comprehension.

**Video data helps physics learning.** Fine-tuning on our PICA-100K dataset yields consistent improvements across multiple dimensions of physical realism. As shown in Appendix A.5, our model consistently produces more physically plausible results, while other models often exhibit unrealistic lighting effects, implausible object deformations, or invalid state changes. Quantitative results in Tab. 3 further support this. For Flux.1-Kontext, our fine-tuned model achieves a +1.71% improvement in overall accuracy over the base model. In addition, it demonstrates better overall physical consistency, improving from 24.57dB to 25.23dB. Notably, we also observe similar gains on Qwen-Image-Edit: fine-tuning on PICA-100K improves overall accuracy from 58.29% to 62.13%, pushing an open-source backbone beyond 60. Overall consistency also improves significantly, increasing from 19.43dB to 21.91dB. These findings suggest that synthetic supervision signals derived from videos can effectively enhance a model's capacity for physics-aware image editing. They also validate the effectiveness of our video-to-image data generation pipeline in capturing diverse and complex physical phenomena. However, we observe a slight drop in global state transition accuracy and causality consistency, possibly due to limitations in directly using first and last frames of a video to represent meaningful state changes. We plan to explore more fine-grained strategies to extract temporal context and leverage intermediate frames.

We also experimented with using real video data to construct an image editing dataset. Following the data pipeline of UniReal (Chen et al., 2025), we employed Miradata (Ju et al., 2024) to generate 400K edited images (MIRA400K) and trained the model under the same settings. However, as shown in Tab. 3, the model trained on MIRA400K performed even worse in overall accuracy. This further demonstrates the efficiency and effectiveness of our proposed data generation pipeline.

## 4.3 VALIDITY OF PICAEVAL

We conduct a human study using Elo ranking to further validate the effectiveness of PICAEval. As shown in Fig. 6, PICAEval achieves higher correlation with human judgments than the baseline. This result demonstrates that our per-case, region-level human annotations and carefully designed

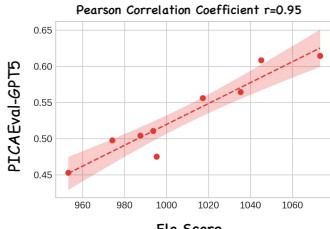 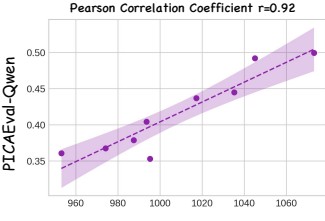 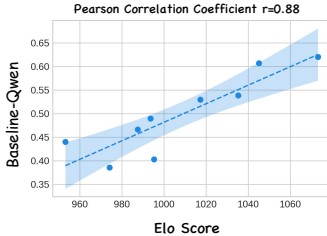

Figure 6: **Alignment between evaluation results and human preference.** We make Pearson correlation analysis between Elo scores from human study and different settings. PICAEval-GPT5, PICAEval-Qwen use GPT-5 and Qwen2.5-VL-72B as the evaluator respectively. Baseline-Qwen adopts Qwen2.5-VL-72B but without edit region annotations. Results show that incorporating stronger VLMs and region-level infomation yields higher alignment with human preference.

questions effectively mitigate VLM hallucinations, leading to outcomes that better reflect human preferences. Additional details of the human study are provided in Appendix A.4.

## 5 LIMITATIONS AND FUTURE DIRECTIONS

While our approach demonstrates clear benefits in physics-aware image editing, it has several limitations. First, the PICA-100K dataset, though effective, is built using a relatively simple generation pipeline and remains limited in scale. Second, our model is trained purely via supervised finetuning (SFT), which brings modest gains but may underexploit the full potential of data. Third, the current framework only supports single-image inputs, lacking the ability to incorporate multi-image or multi-condition contexts. In future work, we aim to enhance the data pipeline, explore RL-based post-training, and extend the model to support more expressive conditioning formats.

## 6 CONCLUSION

We present PICABench, a new benchmark for evaluating physical realism in image editing, along with PICAEval, a region-grounded, QA-based metric for fine-grained assessment. Our results show that current models, still far from producing physically realistic edits. To improve this, we introduce PICA-100K, a synthetic dataset derived from videos. Fine-tuning on this dataset significantly boosts physical consistency, demonstrating the promise of video-based supervision. We hope our benchmark, metric, and dataset can drive progress toward physics-aware image editing.

## 7 ACKNOWLEDGEMENT

This work is supported by Shanghai Artificial Intelligence Laboratory.

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

# A    MORE DETAILS OF PICABENCH

## A.1    TASK DEFINITION

### A.1.1    OPTICS

**Light propagation** requires shadows that are geometrically consistent with the dominant light source, including direction, length, softness, and occlusion. Typical failure modes include misaligned or missing cast shadows and flat shading that ignores occluders.

**Reflection** consistency demands view-dependent behavior for specular highlights and mirror reflections. Mirror images must preserve pose and depth; highlight positions should vary with surface curvature and viewpoint. Failures include "floating" reflections or highlights that remain fixed despite evident shape or view changes.

**Refraction** requires continuous, coherent background distortion through transparent or translucent media. When edited objects involve glass or water, background edges should bend and scale according to interface geometry, with preserved edge continuity. Discontinuous refractive boundaries or inverted distortions indicate violations.

**Light-source effects** evaluate whether new light-introducing edits (like "add a lamp") are consistent with the global illumination context—color casts, shadow penumbra, and brightness falloff should integrate naturally with the scene. Common issues include mismatched color temperatures, overly hard shadows, or inconsistent falloff relative to distance.

### A.1.2    MECHANICS

**Deformation** assesses whether shape changes respect expected material properties. Rigid objects should not bend plastically; elastic deformations should be smooth and bounded. Texture and patterning should warp consistently with geometry rather than tear or duplicate. For instance, changing a chair's height should not collapse its frame or produce rubber-like bending.

**Causality** requires physically plausible contacts and supports under gravity. Edited objects should not float, interpenetrate, or rest in unstable equilibria (e.g., a heavy object balanced on a non-supporting point). Support relations must imply load transfer and stability. Violations include hovering objects, impossible stacking, and intersecting geometries that break solidity.

### A.1.3    STATE TRANSITION

**Global transitions** affect the entire scene (e.g., day-to-night, dry-to-wet, solid-to-molten). Changes must propagate consistently: illumination color and intensity should update across surfaces; wetness should alter reflectance and darkening on all relevant materials; phase changes should be coherent and, when implied, justified by scene-level cues (e.g., a pervasive heat source). Inconsistencies include night skies with daylight shadows or partial melting without corresponding global evidence.

**Local transitions** involve spatially confined edits (e.g., adding steam, charring an edge, or melting a corner). These effects must integrate with nearby context and causal cues. Steam implies heat and moisture and may induce local condensation; flames produce light spill and secondary reflections; partial melting should respect material boundaries and continuity. When localized changes ignore surrounding context or violate material behavior, the edit becomes physically implausible.

## A.2    MORE BENCHMARK RESULTS

Tab. 4 lists the performance of models on PICABench, evaluated by Qwen2.5-VL-72B (Bai et al., 2025). It can be seen that the general rule and conclusion are similar to those suggested by Tab. 1: Most models have very low scores (below 60), indicating a fatal gap in the ability to generate physics-aware images.

We present benchmark results on different prompt levels. The results of PICABench with intermediate prompts evaluated by GPT-5 are shown in Table 5, while the results with explicit prompts evaluated by GPT-5 are provided in Table 6. The results of PICABench using intermediate prompts evaluated by

Table 4: **Quantitative comparison on PICABench-Superficial** evaluated by Qwen2.5-VL-72B for instruction-based editing models, where Acc ↑, Con ↑, LP, LSE, GST, LST denote Accuracy (%) and Consistency (dB), Light propagation, Light Source Effects, Global State Transition, Local State Transition respectively. ■ and ■ indicates the best and second best score in a category, respectively.

| Model | LP | | LSE | | Reflection | | Refraction | | Deformation | | Causality | | GST | | LST | | Overall | |
|---|---|---|---|---|---|---|---|---|---|---|---|---|---|---|---|---|---|---|
| | Acc↑ | Con↑ | Acc↑ | Con↑ | Acc↑ | Con↑ | Acc↑ | Con↑ | Acc↑ | Con↑ | Acc↑ | Con↑ | Acc↑ | Con↑ | Acc↑ | Con↑ | Acc↑ | Con↑ |
| Nano Banana | 48.91 | 27.37 | 47.98 | 28.14 | 51.43 | 25.52 | 52.63 | 25.36 | 44.25 | 24.81 | 44.30 | 25.96 | 52.21 | 13.55 | 50.73 | 24.70 | 49.08 | 23.47 |
| GPT-Image-1 | 58.35 | 16.82 | 56.06 | 15.71 | 50.76 | 17.20 | 63.60 | 17.00 | 40.83 | 17.62 | 46.27 | 17.21 | 63.36 | 10.62 | 52.81 | 15.01 | 53.96 | 15.48 |
| Seedream 4.0 | 55.45 | 25.28 | 57.41 | 27.47 | 55.82 | 24.86 | 54.39 | 26.55 | 45.72 | 24.60 | 47.82 | 26.66 | 59.16 | 11.17 | 56.13 | 28.54 | 54.23 | 23.26 |
| Nano Banana Pro | 52.54 | 27.40 | 57.95 | 23.78 | 55.65 | 27.02 | 54.39 | 27.46 | 47.19 | 25.55 | 50.35 | 26.44 | 65.01 | 11.30 | 58.84 | 23.30 | 56.06 | 22.97 |
| GPT-Image-1.5 | 54.96 | 23.54 | 63.56 | 22.21 | 54.30 | 24.56 | 61.40 | 23.89 | 44.99 | 25.24 | 48.95 | 26.16 | 66.11 | 11.34 | 60.29 | 22.97 | 56.60 | 21.73 |
| DiMOO | 38.26 | 24.08 | 24.80 | 26.68 | 30.19 | 22.12 | 46.05 | 20.76 | 24.94 | 25.53 | 31.50 | 23.19 | 18.21 | 22.56 | 31.19 | 25.44 | 28.57 | 23.70 |
| Uniworld-V1 | 33.41 | 18.50 | 29.11 | 19.96 | 32.88 | 18.59 | 46.49 | 17.48 | 26.16 | 18.82 | 32.91 | 18.11 | 18.54 | 17.62 | 29.94 | 19.41 | 29.18 | 18.48 |
| Bagel | 38.50 | 34.12 | 36.39 | 35.53 | 37.27 | 33.11 | 51.75 | 28.36 | 30.32 | 33.12 | 36.29 | 33.51 | 40.18 | 10.48 | 40.12 | 30.53 | 38.23 | 28.42 |
| Bagel-Think | 39.71 | 32.44 | 43.67 | 29.10 | 36.26 | 33.37 | 48.68 | 28.87 | 33.74 | 27.59 | 36.71 | 32.88 | 39.62 | 11.66 | 39.09 | 27.28 | 38.86 | 26.88 |
| OmniGen2 | 36.32 | 25.69 | 41.78 | 28.34 | 45.36 | 27.78 | 48.25 | 24.84 | 32.52 | 29.28 | 36.29 | 26.93 | 42.72 | 12.18 | 33.89 | 25.89 | 39.52 | 24.12 |
| Hidream-E1.1 | 40.44 | 22.38 | 41.24 | 22.87 | 41.15 | 20.44 | 48.25 | 22.68 | 30.32 | 21.16 | 37.13 | 21.36 | 48.23 | 9.20 | 38.46 | 19.66 | 40.95 | 18.91 |
| Step1X-Edit | 37.29 | 30.38 | 43.94 | 27.53 | 40.64 | 29.32 | 43.42 | 32.37 | 32.03 | 29.71 | 35.72 | 30.92 | 48.79 | 8.75 | 38.46 | 20.92 | 40.59 | 24.68 |
| Flux.1 Kontext | 48.43 | 27.58 | 53.64 | 25.97 | 43.84 | 26.92 | 43.86 | 26.76 | 33.74 | 28.86 | 34.04 | 29.69 | 41.06 | 12.52 | 37.01 | 25.70 | 41.07 | 24.57 |
| Flux.1 Kontext+$SFT$ | 49.64 | 27.90 | 51.21 | 26.58 | 47.22 | 26.99 | 46.49 | 27.01 | 33.99 | 29.19 | 35.44 | 29.54 | 39.29 | 14.44 | 40.75 | 27.01 | 41.93 | 25.23 |
| Δ Improvement | +1.21 | +0.32 | -2.43 | +0.61 | +3.38 | +0.07 | +2.63 | +0.25 | +0.25 | +0.33 | +1.40 | -0.15 | -1.77 | +1.92 | +3.74 | +1.31 | +0.86 | +0.66 |
| Qwen-Image-Edit | 52.54 | 19.87 | 52.02 | 23.07 | 49.07 | 21.56 | 57.46 | 23.72 | 38.14 | 21.49 | 42.62 | 22.65 | 57.73 | 10.19 | 47.82 | 20.26 | 49.71 | 19.43 |
| Qwen-Image-Edit+$SFT$ | 52.89 | 23.24 | 60.47 | 21.91 | 55.19 | 25.33 | 56.12 | 29.73 | 40.99 | 23.66 | 46.24 | 25.51 | 55.25 | 10.57 | 51.27 | 25.60 | 52.06 | 21.91 |
| Δ Improvement | +0.35 | +3.37 | +8.45 | -1.16 | +6.12 | +3.77 | -1.34 | +6.01 | +2.85 | +2.17 | +3.62 | +2.86 | -2.48 | +0.38 | +3.45 | +5.34 | +2.35 | +2.48 |

Table 5: **Quantitative comparison on PICABench-Intermediate** evaluated by GPT-5 for instruction-based editing models, where Acc ↑, Con ↑, LP, LSE, GST, LST denote Accuracy (%) and Consistency (dB), Light propagation, Light Source Effects, Global State Transition, Local State Transition respectively. ■ and ■ indicates the best and second best score in a category, respectively.

| Model | LP | | LSE | | Reflection | | Refraction | | Deformation | | Causality | | GST | | LST | | Overall | |
|---|---|---|---|---|---|---|---|---|---|---|---|---|---|---|---|---|---|---|
| | Acc↑ | Con↑ | Acc↑ | Con↑ | Acc↑ | Con↑ | Acc↑ | Con↑ | Acc↑ | Con↑ | Acc↑ | Con↑ | Acc↑ | Con↑ | Acc↑ | Con↑ | Acc↑ | Con↑ |
| Nano Banana | 56.17 | 27.27 | 63.88 | 26.79 | 61.72 | 25.91 | 53.07 | 26.24 | 63.57 | 23.62 | 63.57 | 23.77 | 66.11 | 13.16 | 64.86 | 25.09 | 62.72 | 22.91 |
| Seedream 4.0 | 63.68 | 21.08 | 70.89 | 18.35 | 68.63 | 22.30 | 50.88 | 24.76 | 63.57 | 22.17 | 63.15 | 21.43 | 71.30 | 10.97 | 68.20 | 26.78 | 65.86 | 20.06 |
| GPT-Image-1 | 66.10 | 16.58 | 74.39 | 14.92 | 64.08 | 16.83 | 62.72 | 16.85 | 64.30 | 16.73 | 63.57 | 16.37 | 78.59 | 10.36 | 69.23 | 16.58 | 68.87 | 15.22 |
| Nano Banana Pro | 64.41 | 26.60 | 74.66 | 21.91 | 70.99 | 26.28 | 67.54 | 26.53 | 69.68 | 26.04 | 68.50 | 24.09 | 76.82 | 11.19 | 70.30 | 26.65 | 70.30 | 22.55 |
| GPT-Image-1.5 | 67.31 | 21.99 | 77.26 | 16.93 | 69.65 | 24.46 | 61.84 | 22.75 | 70.79 | 23.82 | 67.93 | 23.31 | 79.25 | 10.66 | 72.35 | 24.41 | 71.32 | 20.37 |
| DiMOO | 46.25 | 23.86 | 32.61 | 26.85 | 43.00 | 22.55 | 32.02 | 23.38 | 40.34 | 25.12 | 34.32 | 23.09 | 21.30 | 23.75 | 39.09 | 25.55 | 34.78 | 24.09 |
| Uniworld-V1 | 47.22 | 18.71 | 39.35 | 19.44 | 49.75 | 18.75 | 32.89 | 19.52 | 45.97 | 19.39 | 36.99 | 18.39 | 26.27 | 17.33 | 39.71 | 19.24 | 38.69 | 18.62 |
| Bagel | 55.93 | 23.78 | 61.73 | 18.94 | 57.50 | 28.91 | 47.37 | 21.08 | 49.88 | 22.60 | 44.87 | 26.09 | 57.51 | 8.81 | 56.13 | 23.15 | 54.06 | 21.14 |
| Bagel-Think | 48.91 | 28.45 | 55.53 | 21.74 | 50.93 | 31.55 | 50.88 | 24.15 | 53.30 | 26.02 | 42.76 | 30.18 | 53.53 | 9.88 | 52.39 | 25.64 | 50.71 | 24.04 |
| OmniGen2 | 57.38 | 17.15 | 58.22 | 17.63 | 61.38 | 20.94 | 42.98 | 21.34 | 52.32 | 20.64 | 44.02 | 21.41 | 48.79 | 12.74 | 38.05 | 16.78 | 50.27 | 18.19 |
| Hidream-E1.1 | 56.17 | 17.01 | 62.26 | 17.89 | 57.84 | 17.43 | 43.86 | 18.44 | 52.08 | 16.95 | 44.16 | 18.34 | 61.92 | 9.23 | 50.94 | 16.16 | 54.45 | 15.82 |
| Step1X-Edit | 52.06 | 27.83 | 56.06 | 23.59 | 55.99 | 28.79 | 38.60 | 29.83 | 53.06 | 26.95 | 44.02 | 28.61 | 60.26 | 10.24 | 52.39 | 23.85 | 52.80 | 23.76 |
| Flux.1 Kontext | 58.60 | 25.61 | 63.61 | 23.91 | 61.21 | 26.70 | 33.33 | 26.81 | 54.03 | 26.58 | 49.23 | 26.70 | 53.42 | 12.81 | 49.69 | 25.61 | 53.77 | 23.42 |
| Flux.1 Kontext+$SFT$ | 61.26 | 27.23 | 64.15 | 25.73 | 65.43 | 26.73 | 39.47 | 26.54 | 58.92 | 27.41 | 48.10 | 27.81 | 52.76 | 14.38 | 53.22 | 27.16 | 55.59 | 24.53 |
| Δ Improvement | +2.66 | +1.62 | +0.54 | +1.82 | +4.22 | +0.03 | +6.14 | -0.27 | +4.89 | +0.83 | -1.13 | +1.11 | -0.66 | +1.57 | +3.53 | +1.55 | +1.82 | +1.11 |
| Qwen-Image-Edit | 62.47 | 20.37 | 65.23 | 20.28 | 66.61 | 22.49 | 44.74 | 25.54 | 54.77 | 24.26 | 50.49 | 22.36 | 67.44 | 10.91 | 60.91 | 22.78 | 60.41 | 20.14 |
| Qwen-Image-Edit+$SFT$ | 63.40 | 21.70 | 69.67 | 18.96 | 70.38 | 24.92 | 54.94 | 28.74 | 59.61 | 24.43 | 56.81 | 23.66 | 65.71 | 10.93 | 61.74 | 26.70 | 63.25 | 21.31 |
| Δ Improvement | +0.93 | +1.33 | +4.44 | -1.32 | +3.77 | +2.43 | +10.20 | +3.20 | +4.84 | +0.17 | +6.32 | +1.30 | -1.73 | +0.02 | +0.83 | +3.92 | +2.84 | +1.17 |

Qwen2.5-VL-72B is shown in Table 7, and the results of PICABench with explicit prompts evaluated by GPT-5 are reported in Table 8.

As demonstrated above, model performance improves with more informative prompts. Furthermore, the fine-tuned model consistently outperforms the baseline, indicating the effectiveness of the PICA100K dataset.

## A.3 Details of Benchmark Metrics

We provide detailed definition of accuracy and consistency as follows. Let $N$ be the total number of annotated QA pairs, $\hat{a}_i$ be the VLM-predicted answer for the $i$-th question, $a_i$ be the reference answer, and $\mathbb{I}(\cdot)$ the indicator function. The accuracy is defined as:

$$\text{Acc} = \frac{1}{N} \sum_{i=1}^{N} \mathbb{I}(\hat{a}_i = a_i) \quad (1)$$

Table 6: **Quantitative comparison on PICABench-Explicit** evaluated by GPT-5 for instruction-based editing models, where Acc ↑, Con ↑, LP, LSE, GST, LST denote Accuracy (%) and Consistency (dB), Light propagation, Light Source Effects, Global State Transition, Local State Transition respectively. ■ and ■ indicates the best and second best score in a category, respectively.

| Model | LP | | LSE | | Reflection | | Refraction | | Deformation | | Causality | | GST | | LST | | Overall | |
|---|---|---|---|---|---|---|---|---|---|---|---|---|---|---|---|---|---|---|
| | Acc ↑ | Con ↑ | Acc ↑ | Con ↑ | Acc ↑ | Con ↑ | Acc ↑ | Con ↑ | Acc ↑ | Con ↑ | Acc ↑ | Con ↑ | Acc ↑ | Con ↑ | Acc ↑ | Con ↑ | Acc ↑ | Con ↑ |
| Nano Banana | 63.92 | 26.98 | 65.23 | 25.89 | 68.30 | 25.12 | 54.39 | 25.79 | 66.99 | 24.41 | 68.21 | 24.87 | 69.32 | 13.37 | 64.66 | 25.50 | 66.46 | 23.02 |
| GPT-Image-1 | 63.92 | 16.34 | 72.51 | 14.97 | 66.44 | 16.46 | 60.53 | 16.73 | 63.08 | 16.87 | 63.85 | 16.40 | 78.48 | 10.46 | 64.66 | 16.44 | 68.07 | 15.16 |
| Nano Banana Pro | 65.13 | 26.75 | 74.12 | 22.26 | 68.80 | 26.27 | 62.28 | 27.12 | 71.64 | 24.98 | 74.82 | 25.08 | 81.46 | 11.74 | 73.80 | 26.12 | 72.69 | 22.75 |
| Seedream 4.0 | 65.62 | 19.87 | 73.32 | 16.76 | 69.14 | 20.42 | 60.53 | 24.77 | 69.93 | 20.65 | 76.23 | 20.58 | 82.23 | 9.69 | 76.72 | 24.95 | 73.76 | 18.73 |
| GPT-Image-1.5 | 67.31 | 21.23 | 83.56 | 16.38 | 69.31 | 23.25 | 65.79 | 22.33 | 73.76 | 22.25 | 77.07 | 22.94 | 84.66 | 10.55 | 78.17 | 23.01 | 76.06 | 19.62 |
| DiMOO | 44.55 | 24.72 | 29.92 | 28.06 | 40.47 | 23.17 | 26.32 | 23.56 | 47.43 | 25.25 | 33.19 | 24.35 | 20.86 | 23.74 | 41.58 | 25.66 | 34.39 | 24.65 |
| Uniworld-V1 | 47.70 | 17.95 | 42.86 | 19.03 | 51.77 | 18.34 | 31.58 | 19.06 | 49.39 | 18.76 | 38.82 | 18.17 | 35.76 | 15.78 | 46.15 | 18.93 | 42.78 | 17.96 |
| Bagel | 62.71 | 15.39 | 72.24 | 13.89 | 62.39 | 18.74 | 55.26 | 16.68 | 57.70 | 15.24 | 59.35 | 19.98 | 77.26 | 8.14 | 65.90 | 15.65 | 65.61 | 15.20 |
| Bagel-Think | 52.54 | 27.05 | 60.38 | 17.96 | 57.84 | 29.35 | 43.86 | 24.63 | 55.99 | 24.64 | 48.10 | 28.01 | 62.91 | 9.34 | 57.80 | 23.38 | 56.01 | 22.34 |
| OmniGen2 | 57.14 | 12.81 | 57.14 | 14.62 | 64.59 | 17.54 | 46.93 | 18.02 | 56.72 | 15.91 | 55.98 | 18.88 | 59.82 | 10.39 | 51.98 | 15.38 | 57.39 | 15.19 |
| Hidream-E1.1 | 57.87 | 14.76 | 68.73 | 15.46 | 61.21 | 15.52 | 46.49 | 15.91 | 64.06 | 15.98 | 55.70 | 16.00 | 71.30 | 9.55 | 60.71 | 15.52 | 62.23 | 14.40 |
| Step1X-Edit | 55.45 | 22.26 | 62.26 | 18.86 | 61.55 | 25.58 | 40.79 | 26.99 | 58.92 | 25.12 | 56.82 | 26.01 | 67.44 | 9.42 | 58.63 | 23.52 | 59.73 | 21.25 |
| Flux.1 Kontext | 61.74 | 25.61 | 68.19 | 21.20 | 63.24 | 25.89 | 42.98 | 24.89 | 58.44 | 24.75 | 62.59 | 23.54 | 70.75 | 10.70 | 61.75 | 23.68 | 63.30 | 21.54 |
| Flux.1 Kontext+*SFT* | 62.95 | 26.82 | 69.00 | 23.04 | 63.91 | 26.07 | 46.05 | 25.68 | 63.81 | 26.47 | 66.10 | 25.15 | 67.99 | 11.08 | 63.20 | 25.19 | 64.47 | 22.63 |
| Δ Improvement | +1.21 | +1.21 | +0.81 | +1.84 | +0.67 | +0.18 | +3.07 | +0.79 | +5.37 | +1.72 | +3.51 | +1.61 | -2.76 | +0.38 | +1.45 | +1.51 | +1.17 | +1.09 |
| Qwen-Image-Edit | 65.62 | 17.78 | 71.43 | 18.81 | 64.59 | 20.45 | 51.75 | 24.29 | 58.44 | 21.14 | 66.39 | 19.90 | 78.81 | 9.76 | 69.65 | 19.33 | 68.02 | 17.96 |
| Qwen-Image-Edit+*SFT* | 64.07 | 20.16 | 77.86 | 16.90 | 69.57 | 22.31 | 50.73 | 24.88 | 69.32 | 20.25 | 69.90 | 20.27 | 76.16 | 10.03 | 69.23 | 24.44 | 70.05 | 18.88 |
| Δ Improvement | -1.55 | +2.38 | +6.43 | -1.91 | +4.98 | +1.86 | -1.02 | +0.59 | +10.88 | -0.89 | +3.51 | +0.37 | -2.65 | +0.27 | -0.42 | +5.11 | +2.03 | +0.92 |

Table 7: **Quantitative comparison on PICABench-Intermediate** evaluated by Qwen2.5-VL-72B for instruction-based editing models, where Acc ↑, Con ↑, LP, LSE, GST, LST denote Accuracy (%) and Consistency (dB), Light propagation, Light Source Effects, Global State Transition, Local State Transition respectively. ■ and ■ indicates the best and second best score in a category, respectively.

| Model | LP | | LSE | | Reflection | | Refraction | | Deformation | | Causality | | GST | | LST | | Overall | |
|---|---|---|---|---|---|---|---|---|---|---|---|---|---|---|---|---|---|---|
| | Acc ↑ | Con ↑ | Acc ↑ | Con ↑ | Acc ↑ | Con ↑ | Acc ↑ | Con ↑ | Acc ↑ | Con ↑ | Acc ↑ | Con ↑ | Acc ↑ | Con ↑ | Acc ↑ | Con ↑ | Acc ↑ | Con ↑ |
| Nano Banana | 48.67 | 27.27 | 50.67 | 26.79 | 50.93 | 25.91 | 49.12 | 26.24 | 43.77 | 23.62 | 51.34 | 23.77 | 56.84 | 13.16 | 53.22 | 25.09 | 51.51 | 22.91 |
| GPT-Image-1 | 58.35 | 16.58 | 60.92 | 14.92 | 51.10 | 16.83 | 59.21 | 16.85 | 45.72 | 16.73 | 54.15 | 16.37 | 70.09 | 10.36 | 53.85 | 16.58 | 57.66 | 15.22 |
| Seedream 4.0 | 57.14 | 21.08 | 61.73 | 18.35 | 56.66 | 22.30 | 55.70 | 24.76 | 51.10 | 22.17 | 55.56 | 21.43 | 66.00 | 10.97 | 55.09 | 26.78 | 58.24 | 20.06 |
| Nano Banana Pro | 58.60 | 26.60 | 60.65 | 21.91 | 56.66 | 26.28 | 59.65 | 26.53 | 49.88 | 26.04 | 57.38 | 24.09 | 67.88 | 11.19 | 59.46 | 26.65 | 59.21 | 22.55 |
| GPT-Image-1.5 | 58.60 | 21.99 | 66.30 | 16.93 | 56.49 | 24.46 | 57.02 | 22.75 | 51.49 | 23.82 | 58.37 | 23.31 | 71.08 | 10.66 | 60.91 | 24.41 | 60.63 | 20.37 |
| DiMOO | 37.53 | 23.86 | 23.18 | 26.85 | 31.87 | 22.55 | 38.60 | 23.38 | 26.41 | 25.12 | 31.08 | 23.09 | 17.66 | 23.75 | 29.52 | 25.55 | 27.94 | 24.09 |
| Uniworld-V1 | 39.23 | 18.71 | 31.81 | 19.44 | 33.22 | 18.75 | 31.14 | 19.52 | 29.58 | 19.39 | 30.38 | 18.39 | 19.65 | 17.33 | 33.68 | 19.24 | 29.79 | 18.62 |
| Bagel | 48.91 | 23.78 | 55.26 | 18.94 | 46.04 | 28.91 | 56.14 | 21.08 | 37.41 | 22.60 | 36.85 | 26.09 | 48.01 | 8.81 | 44.28 | 23.15 | 45.50 | 21.14 |
| Bagel-Think | 42.37 | 28.45 | 47.71 | 21.74 | 40.81 | 31.55 | 53.95 | 24.15 | 39.12 | 26.02 | 37.55 | 30.18 | 45.70 | 9.88 | 44.49 | 25.64 | 43.09 | 24.04 |
| OmniGen2 | 47.70 | 17.15 | 49.60 | 17.63 | 46.21 | 20.94 | 44.74 | 21.34 | 35.70 | 20.64 | 38.12 | 21.41 | 39.40 | 12.74 | 31.39 | 16.78 | 40.90 | 18.19 |
| Hidream-E1.1 | 46.73 | 17.01 | 53.10 | 17.89 | 46.04 | 17.43 | 48.68 | 18.44 | 38.39 | 16.95 | 38.40 | 18.34 | 56.40 | 9.23 | 41.16 | 16.16 | 46.52 | 15.82 |
| Step1X-Edit | 42.13 | 27.83 | 50.13 | 23.59 | 43.17 | 28.79 | 46.05 | 29.83 | 36.43 | 26.95 | 39.52 | 28.61 | 52.76 | 10.24 | 43.66 | 23.85 | 44.72 | 23.76 |
| Flux.1 Kontext | 48.91 | 25.61 | 55.53 | 23.91 | 45.87 | 26.55 | 43.86 | 26.81 | 38.14 | 26.58 | 44.30 | 26.70 | 44.15 | 12.81 | 43.87 | 25.61 | 45.28 | 23.42 |
| Flux.1 Kontext+*SFT* | 47.70 | 27.23 | 58.22 | 25.73 | 48.74 | 26.73 | 46.49 | 26.54 | 41.56 | 27.41 | 42.33 | 27.81 | 42.27 | 14.38 | 44.49 | 27.16 | 45.62 | 24.53 |
| Δ Improvement | -1.21 | +1.62 | +2.69 | +1.82 | +2.87 | +0.03 | +2.63 | -0.27 | +3.42 | +0.83 | -1.97 | +1.11 | -1.88 | +1.57 | +0.62 | +1.55 | +0.34 | +1.11 |
| Qwen-Image-Edit | 54.24 | 20.37 | 58.49 | 20.28 | 50.42 | 22.49 | 49.12 | 25.54 | 42.30 | 24.26 | 43.46 | 22.36 | 57.40 | 10.91 | 50.31 | 22.78 | 50.97 | 20.14 |
| Qwen-Image-Edit+*SFT* | 56.59 | 21.70 | 58.89 | 18.96 | 54.41 | 24.92 | 53.73 | 28.74 | 43.62 | 24.43 | 50.01 | 23.66 | 54.93 | 10.93 | 51.20 | 26.70 | 52.88 | 21.31 |
| Δ Improvement | +2.35 | +1.33 | +0.40 | -1.32 | +3.99 | +2.43 | +4.61 | +3.20 | +1.32 | +0.17 | +6.55 | +1.30 | -2.47 | +0.02 | +0.89 | +3.92 | +1.91 | +1.17 |

We use psnr of non-edited region as consistency. For each image pair, we compute the PSNR over non-edited pixels, using a binary mask $M_i(p)$ where $M_i(p) = 1$ denotes an edited pixel and $M_i(p) = 0$ otherwise. Define the non-edited region as $\Omega_i = \{p \mid M_i(p) = 0\}$, where $p$ indexes pixels. Let $I_i^{\text{src}}$ be the source image and $I_i^{\text{edit}}$ the edited image.

The MSE (mean squared error) over the non-edited region is:

$$\text{MSE}_i = \frac{1}{|\Omega_i|} \sum_{p \in \Omega_i} \left\| I_i^{\text{edit}}(p) - I_i^{\text{src}}(p) \right\|_2^2 \tag{2}$$

Then, the per-sample consistency score (PSNR) is:

$$\text{Con}_i = 10 \cdot \log_{10} \left( \frac{\text{MAX}^2}{\text{MSE}_i} \right) \tag{3}$$

Table 8: **Quantitative comparison on PICABench-Explicit** evaluated by Qwen2.5-VL-72B for instruction-based editing models, where Acc ↑, Con ↑, LP, LSE, GST, LST denote Accuracy (%) and Consistency (dB), Light propagation, Light Source Effects, Global State Transition, Local State Transition respectively. ■ and ■ indicates the best and second best score in a category, respectively.

| Model | LP | | LSE | | Reflection | | Refraction | | Deformation | | Causality | | GST | | LST | | Overall | |
|---|---|---|---|---|---|---|---|---|---|---|---|---|---|---|---|---|---|---|
| | Acc ↑ | Con ↑ | Acc ↑ | Con ↑ | Acc ↑ | Con ↑ | Acc ↑ | Con ↑ | Acc ↑ | Con ↑ | Acc ↑ | Con ↑ | Acc ↑ | Con ↑ | Acc ↑ | Con ↑ | Acc ↑ | Con ↑ |
| Nano Banana | 53.27 | 26.98 | 54.45 | 25.89 | 55.99 | 25.12 | 56.58 | 25.79 | 47.68 | 24.41 | 58.93 | 24.87 | 58.17 | 13.37 | 52.81 | 25.50 | 55.40 | 23.02 |
| GPT-Image-1 | 59.56 | 16.34 | 61.99 | 14.97 | 52.61 | 16.46 | 61.84 | 16.73 | 44.99 | 16.87 | 53.16 | 16.40 | 70.53 | 10.46 | 51.56 | 16.44 | 57.83 | 15.16 |
| Nano Banana Pro | 59.32 | 26.75 | 64.69 | 22.26 | 61.38 | 26.27 | 60.09 | 27.12 | 53.55 | 24.98 | 64.70 | 25.08 | 72.08 | 11.74 | 63.41 | 26.12 | 63.29 | 22.75 |
| Seedream 4.0 | 58.84 | 19.87 | 66.04 | 16.76 | 58.85 | 20.42 | 62.72 | 24.77 | 50.12 | 20.65 | 67.09 | 20.58 | 77.37 | 9.69 | 63.62 | 24.95 | 64.91 | 18.73 |
| GPT-Image-1.5 | 62.95 | 21.23 | 71.43 | 16.38 | 61.21 | 23.25 | 62.28 | 22.33 | 57.18 | 22.25 | 67.23 | 22.94 | 76.93 | 10.55 | 66.11 | 23.01 | 67.01 | 19.62 |
| DiMOO | 32.93 | 24.72 | 23.99 | 28.06 | 27.82 | 23.17 | 31.14 | 23.56 | 30.32 | 25.25 | 27.29 | 24.35 | 17.99 | 23.74 | 33.06 | 25.66 | 26.78 | 24.65 |
| Uniworld-V1 | 37.77 | 17.95 | 34.50 | 19.03 | 37.44 | 18.34 | 30.70 | 19.06 | 30.32 | 18.76 | 34.18 | 18.17 | 28.81 | 15.78 | 38.67 | 18.93 | 33.80 | 17.96 |
| Bagel | 54.48 | 15.39 | 63.34 | 13.89 | 52.28 | 18.74 | 55.70 | 16.68 | 42.05 | 15.24 | 52.32 | 19.98 | 68.43 | 8.14 | 54.05 | 15.65 | 56.44 | 15.20 |
| Bagel-Think | 42.86 | 27.05 | 52.29 | 17.96 | 43.17 | 29.35 | 48.25 | 24.63 | 40.10 | 24.64 | 38.40 | 28.01 | 53.86 | 9.34 | 46.99 | 23.38 | 45.91 | 22.34 |
| OmniGen2 | 51.09 | 12.81 | 47.98 | 14.62 | 48.74 | 17.54 | 45.18 | 18.02 | 42.79 | 15.91 | 48.24 | 18.88 | 52.76 | 10.39 | 42.41 | 15.38 | 48.18 | 15.19 |
| Hidream-E1.1 | 49.39 | 14.76 | 54.99 | 15.46 | 44.52 | 15.52 | 45.18 | 15.91 | 40.83 | 15.98 | 45.15 | 16.00 | 58.72 | 9.55 | 47.61 | 15.52 | 49.22 | 14.40 |
| Step1X-Edit | 43.10 | 22.26 | 52.29 | 18.86 | 47.05 | 25.58 | 47.37 | 26.99 | 40.34 | 25.12 | 46.69 | 26.01 | 57.95 | 9.42 | 47.19 | 23.52 | 48.83 | 21.25 |
| Flux.1 Kontext | 52.54 | 25.61 | 59.57 | 21.20 | 51.10 | 25.89 | 46.05 | 24.89 | 40.59 | 24.75 | 53.16 | 23.54 | 62.03 | 10.70 | 54.47 | 23.68 | 53.84 | 21.54 |
| Flux.1 Kontext+$SFT$ | 52.06 | 26.82 | 59.30 | 23.04 | 52.78 | 26.07 | 47.37 | 25.68 | 42.79 | 26.47 | 53.73 | 25.15 | 61.48 | 11.08 | 53.64 | 25.19 | 54.18 | 22.63 |
| Δ Improvement | -0.48 | +1.21 | -0.27 | +1.84 | +1.68 | +0.18 | +1.32 | +0.79 | +2.20 | +1.72 | +0.57 | +1.61 | -0.55 | +0.38 | -0.83 | +1.51 | +0.34 | +1.09 |
| Qwen-Image-Edit | 54.00 | 17.78 | 60.38 | 18.81 | 52.28 | 20.45 | 53.07 | 24.29 | 44.74 | 21.14 | 56.68 | 19.90 | 65.78 | 9.76 | 59.04 | 19.33 | 57.00 | 17.96 |
| Qwen-Image-Edit+$SFT$ | 54.69 | 20.16 | 63.94 | 16.90 | 56.34 | 22.31 | 54.61 | 24.88 | 48.61 | 20.25 | 56.77 | 20.27 | 64.32 | 10.03 | 60.35 | 24.44 | 58.21 | 18.88 |
| Δ Improvement | +0.69 | +2.38 | +3.56 | -1.91 | +4.06 | +1.86 | +1.54 | +0.59 | +3.87 | -0.89 | +0.09 | +0.37 | -1.46 | +0.27 | +1.31 | +5.11 | +1.21 | +0.92 |

Finally, the dataset-level consistency is computed as the average across all $N$ samples:

$$\text{Con} = \frac{1}{N} \sum_{i=1}^{N} \text{Con}_i \qquad (4)$$

Here, $\text{MAX}$ is the maximum pixel value (e.g., 255 for 8-bit images).

### A.4 DETAILED HUMAN EVALUATION PROTOCOL

**Study setup.** We use the Rapidata[1] platform to conduct pairwise human preference comparisons for evaluating image editing quality. Each trial presents a reference image and two model outputs (A/B) under a fixed unified instruction:

> Select the image that more closely matches the editing instruction.

The A/B order is randomized per trial. Annotators are vetted beforehand and participate on a voluntary basis.

**Datasets and models.** We evaluate 9 models over the PICABench dataset at three difficulty levels (*superficial*, *intermediate*, *explicit*), forming 36 unordered model pairs per item. For each difficulty, we sample 50 items via stratified sampling over the `physics_law` taxonomy. Each item yields 36 comparisons, each judged by 5 annotators, resulting in 27,000 votes per split.

**Elo computation.** To aggregate preferences, we use a robust Elo rating system. For a match between model $A$ and $B$ with current ratings $(R_A, R_B)$, the expected win probability of $A$ is:

$$E_A = \frac{1}{1 + 10^{\frac{R_B - R_A}{S}}}, \qquad (5)$$

where $S = 400$ is the scaling factor.

Given the vote ratio $s_A \in [0, 1]$ for model $A$, with $s_B = 1 - s_A$, the ratings are updated as:

$$\begin{aligned} R'_A &= \max(R_{\min}, \ R_A + K_{\text{eff}}(s_A - E_A)), \\ R'_B &= \max(R_{\min}, \ R_B + K_{\text{eff}}(s_B - E_B)), \end{aligned} \qquad (6)$$

where $K_{\text{eff}} = K \cdot \frac{v}{5}$ adjusts for vote count $v = v_A + v_B$, and $K = 24$ is the base step size.

---

[1]https://www.rapidata.ai/

**Robust aggregation.** To reduce order effects and improve stability, we shuffle the comparison stream and re-run Elo updates for $T = 50$ rounds. The final Elo score for model $m$ is computed as:

$$\bar{R}_m = \frac{1}{T} \sum_{t=1}^{T} R_m^{(t)}, \quad \sigma_m = \sqrt{\frac{1}{T} \sum_{t=1}^{T} \left( R_m^{(t)} - \bar{R}_m \right)^2}. \tag{7}$$

**Parameter setting.** Table 9 summarizes the Elo configuration used in all human evaluations.

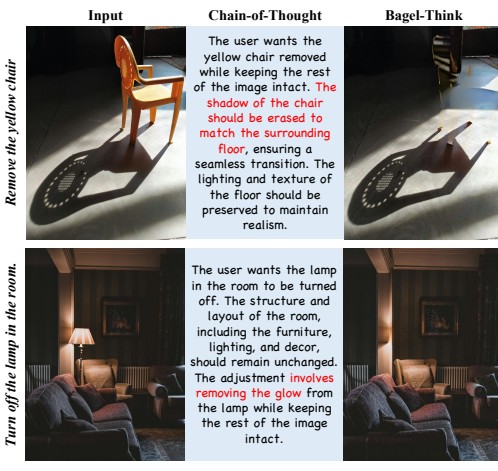

Figure 7: Examples of Bagel's reasoning trace.

Table 9: Elo parameter setting.

| Parameter | Value |
|---|---|
| Initial Elo rating | 1,000 |
| Elo scaling factor $S$ | 400 |
| Base K-factor | 24 |
| Minimum Elo rating $R_{\min}$ | 700 |
| Number of rounds $T$ | 50 |
| Votes per match | 3 |
| Model pairs per item | 45 |
| Items per difficulty | 50 |
| Benchmark splits | 3 |
| Total comparisons per split | 6,750 |
| Total votes per split | 20,250 |

### A.5 MORE VISUALIZATION

Fig. 8-15 presents generated images of various models prompted by samples in our PICABench, where "Ours" indicates our fine-tuned Flux.1-Kontext model. The prompts cover all eight physics laws and three complexity levels. They demonstrate that the performance of these models varies considerably in complying with physical laws. Most models either just perform superficial edits and ignore the physics law, or completely fail to understand the instruction. Only a few models, including ours, can yield physically plausible images in most cases. Therefore, the ability to follow physical laws is crucial but lacking in most models, and by PICABench we hope to draw the community's attention to this critical problem.

Moreover, we show Bagel's think process in Fig. 7. As shown in Fig. 7, model successfully reasons the correct results in its chain-of-thought, yet fails to execute them in the generated image.

### A.6 PROMPTS FOR PICABENCH

We present the prompts used in constructing PICABench. Fig. 17 shows the prompt used to generate QAs to evaluate editing models' performance. Fig. 16 shows the prompts used to generate edit instructions in PICA-100K.

## B MORE DETAILS ABOUT PICA-100K

### B.1 EXAMPLE OF PICA-100K

Fig. 18 shows some examples in PICA-100K dataset. For each pair, we focus on the manifestation of physical laws.

**Superficial Prompt:** Remove the yellow chair

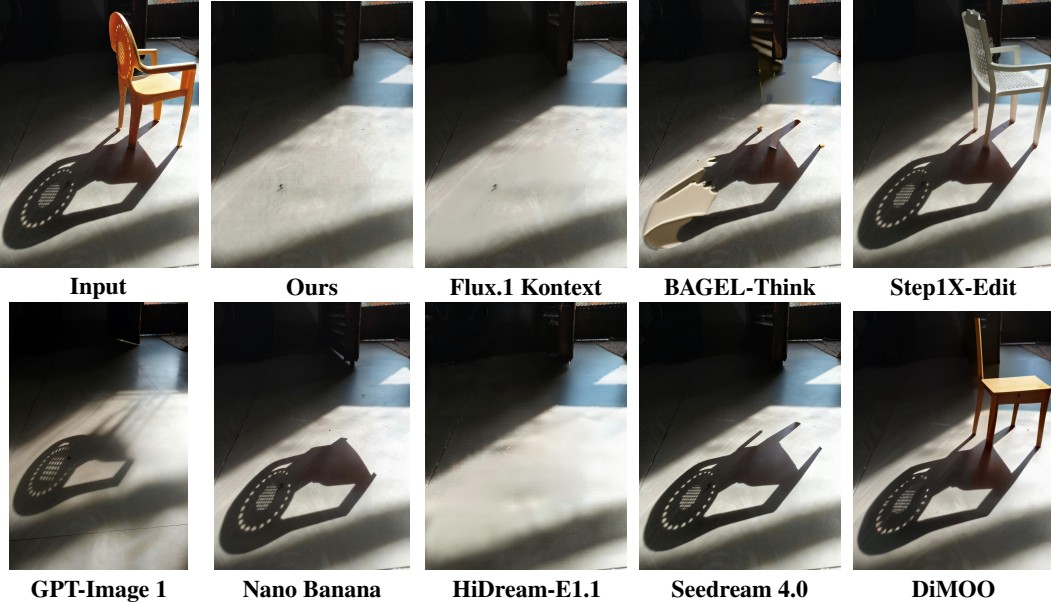

**Superficial Prompt:** Move the potted plant to left side of the table.

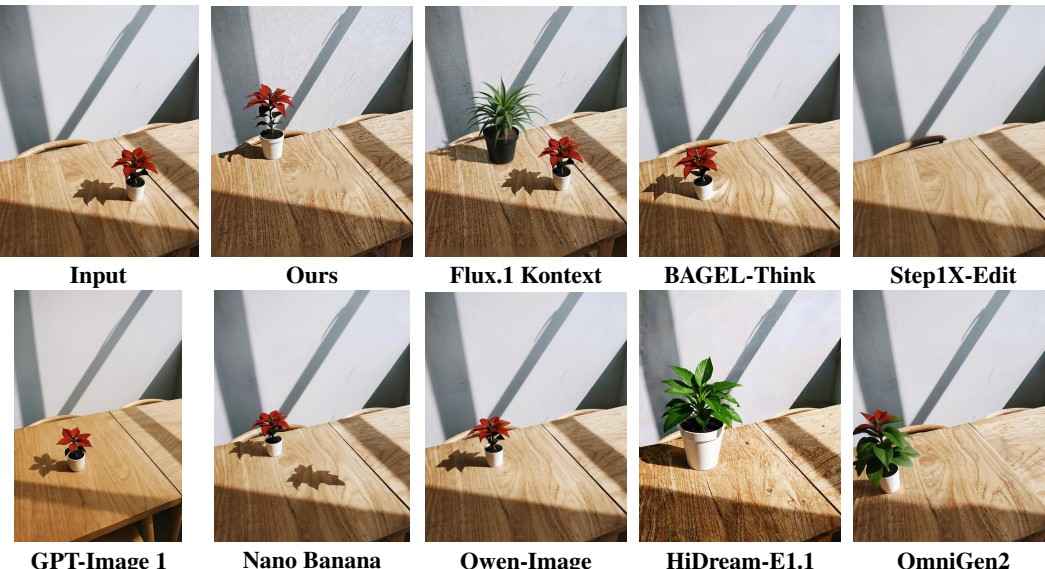

Figure 8: Examples of how models follow the law of light propagation in optics (superficial propmts).

## B.2 PROMPTS FOR PICA-100K

We show the prompts used in constructing PICA-100K. Fig. 19 shows the prompt used in generating video from generated images. Fig. 20 shows the prompts used to generate edit instructions in PICA-100K.

**Superficial Prompt:** Turn on the lamp on the bedside table.

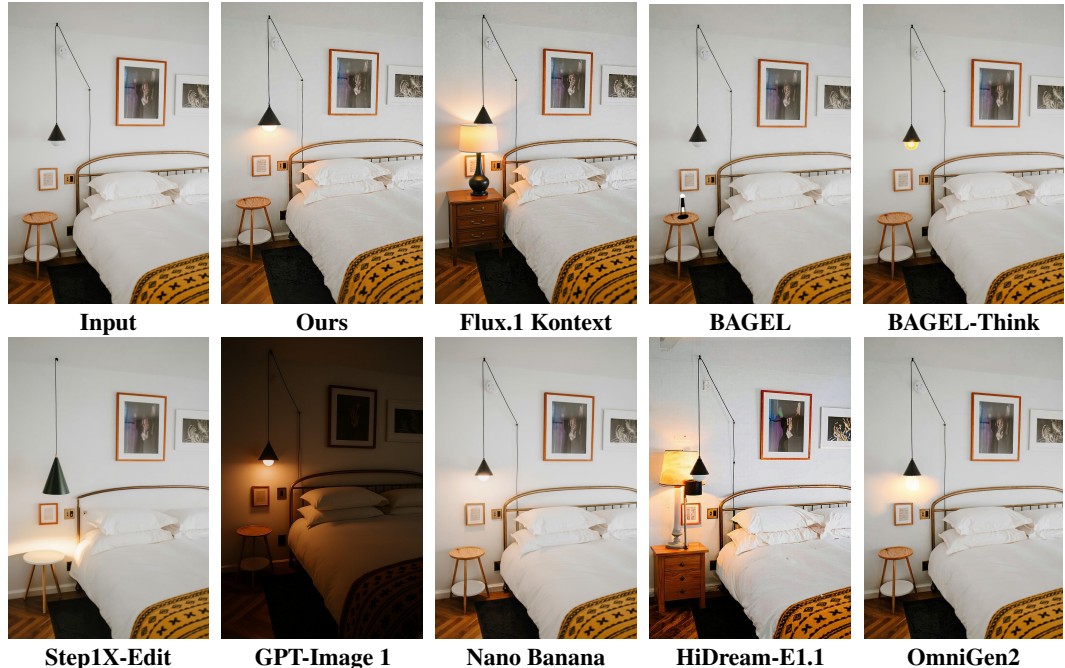

**Superficial Prompt:** Turn off the lamp in the room.

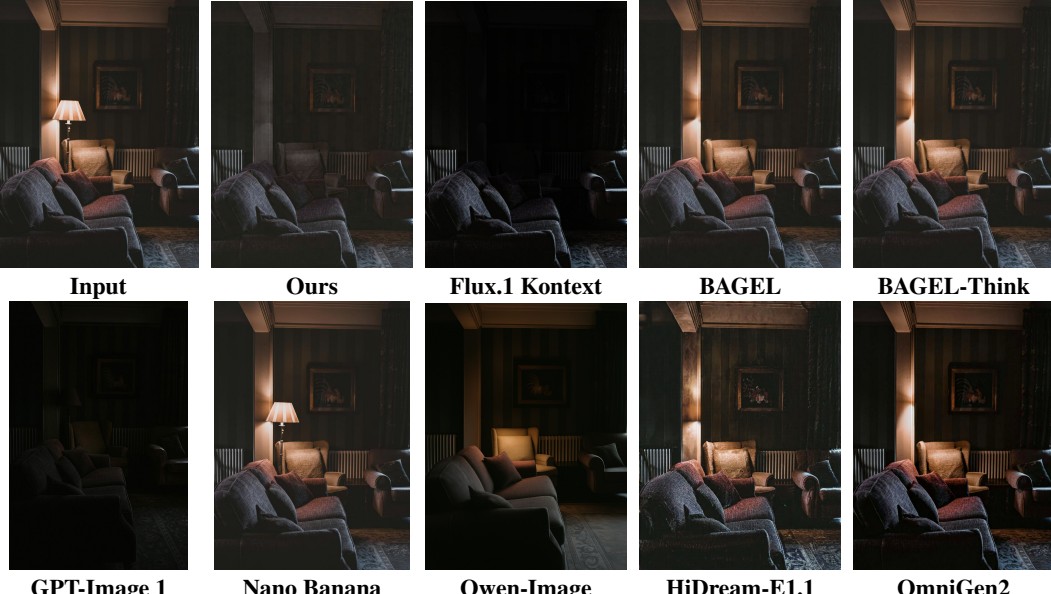

Figure 9: Examples of how models follow the law of light source effects in optics (superficial propmts).

**Superficial Prompt:** Move the hot air balloon in the image to the left.

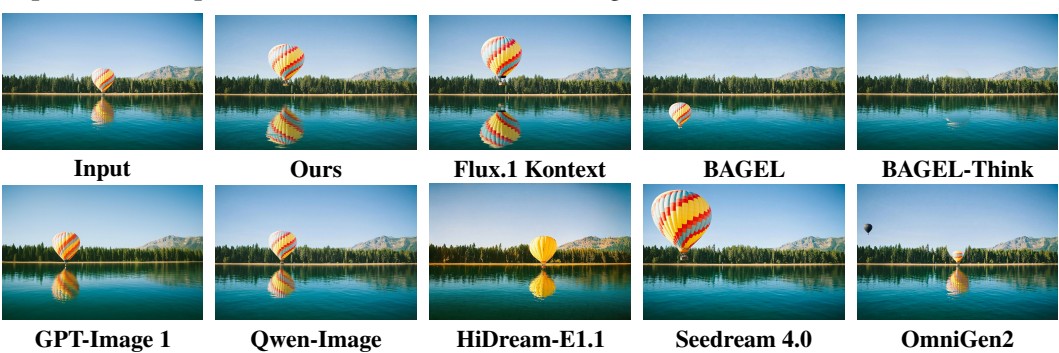

**Superficial Prompt:** Move the man to the right.

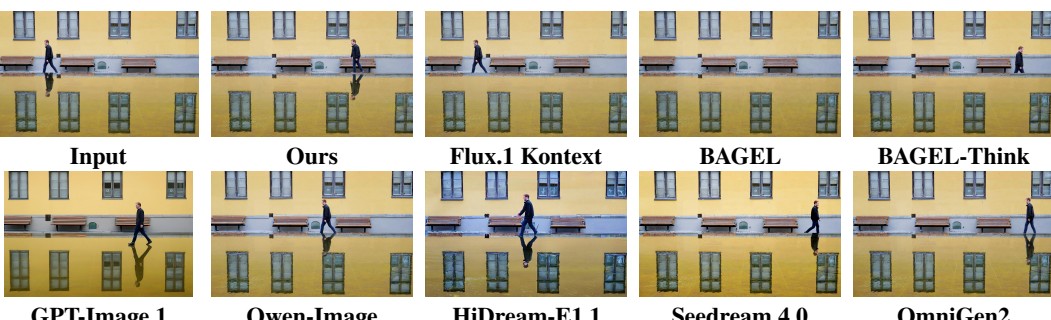

Figure 10: Examples of how models follow the law of reflection in optics (superficial propmts).

**Superficial Prompt:** Remove the magnifying glass from the image.

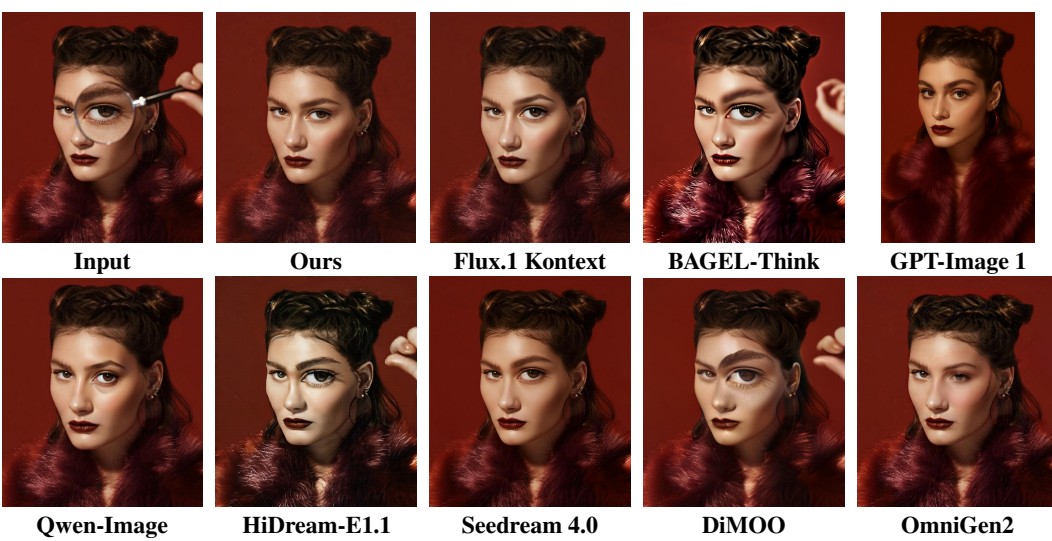

**Superficial Prompt:** Add a blue straw to the glass of water.

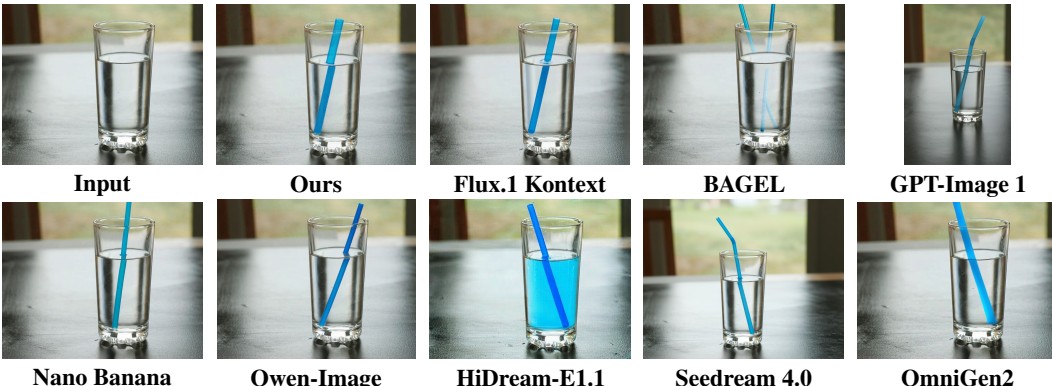

Figure 11: Examples of how models follow the law of refraction in optics (superficial propmts).

**Superficial Prompt:** Add a dog on top of the pillow.

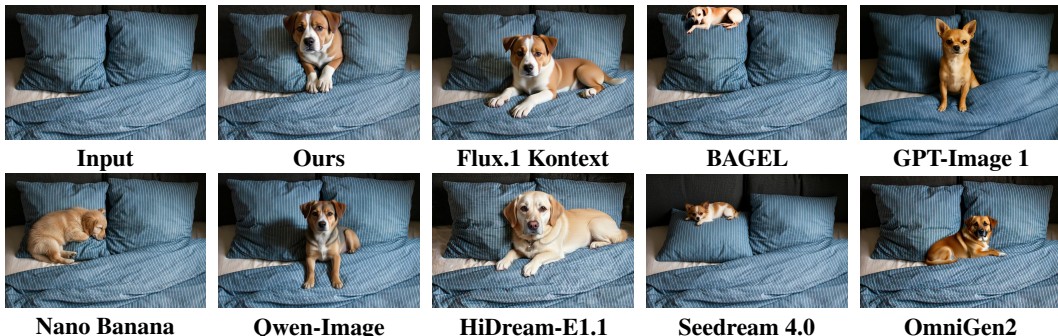

**Superficial Prompt:** Change the woman's position from sitting to standing.

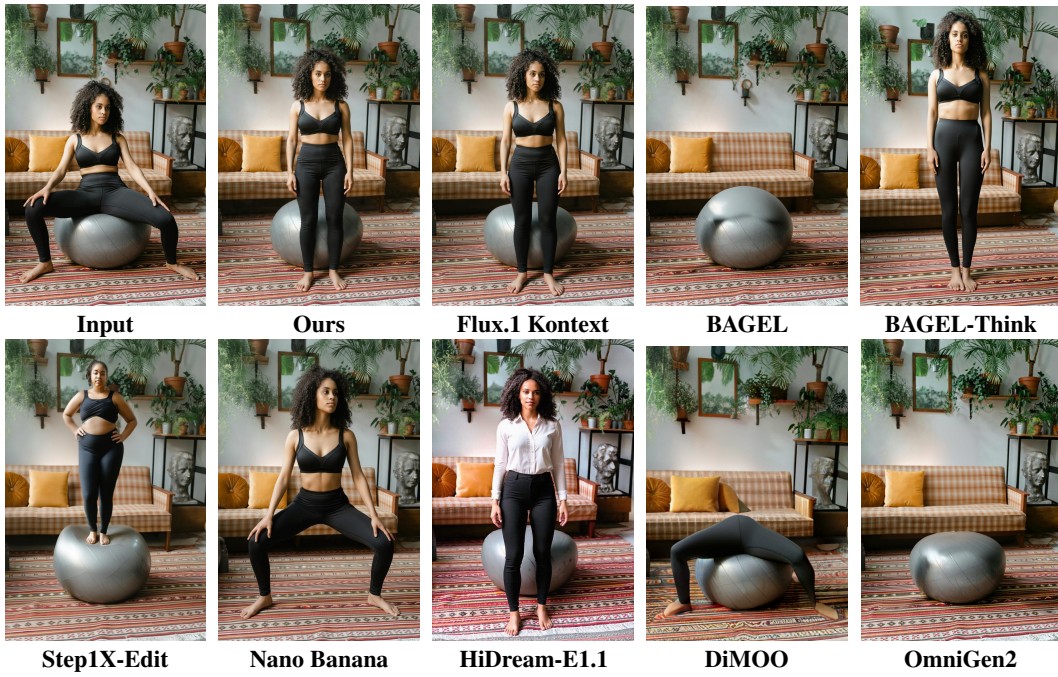

Figure 12: Examples of how models follow the law of deformation in mechanics (superficial propmts).

**Superficial Prompt:** Remove the skateboard.

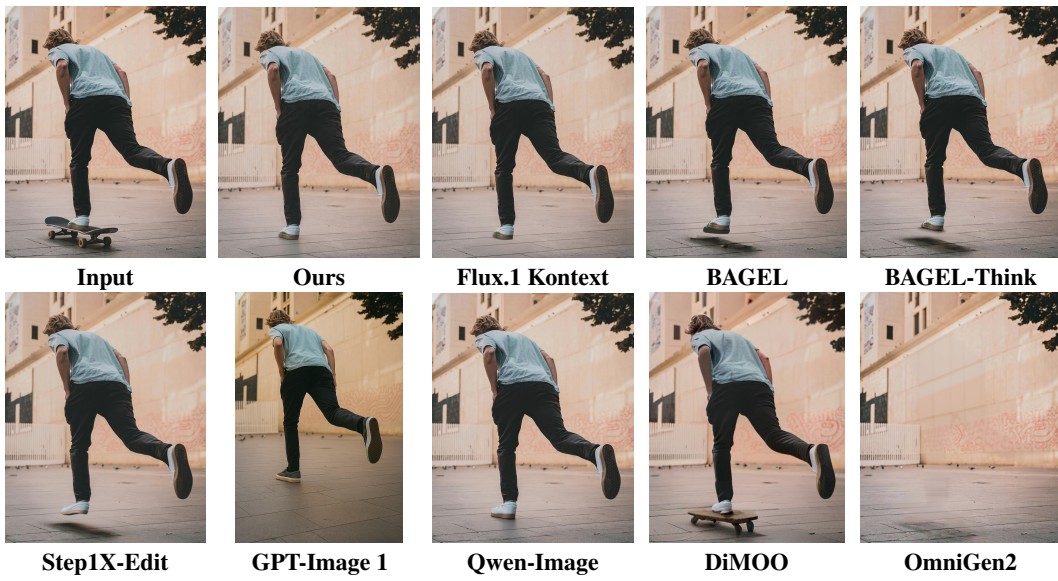

**Superficial Prompt:** Remove the white table.

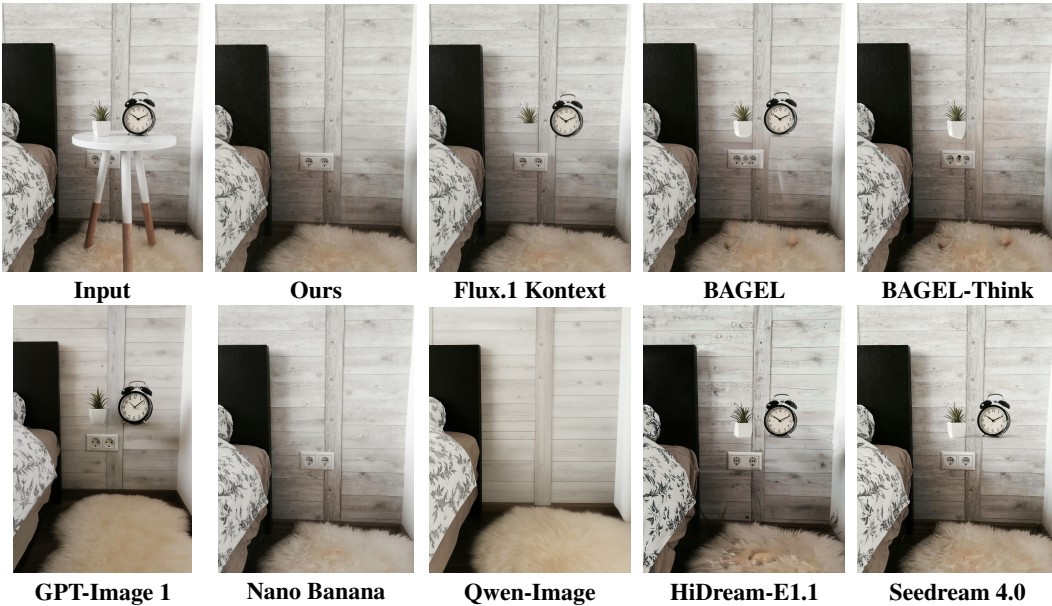

Figure 13: Examples of how models follow the law of causality in mechanics (superficial propmts).

**Superficial Prompt:** Change the foggy weather to a sunny day.

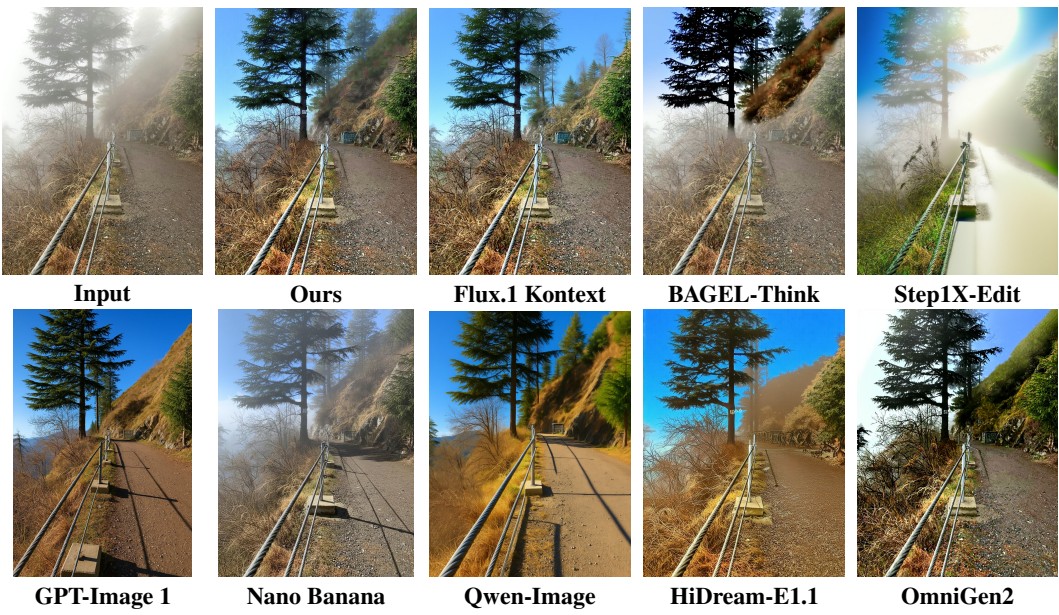

**Superficial Prompt:** Change the weather in the scene to a snowy day.

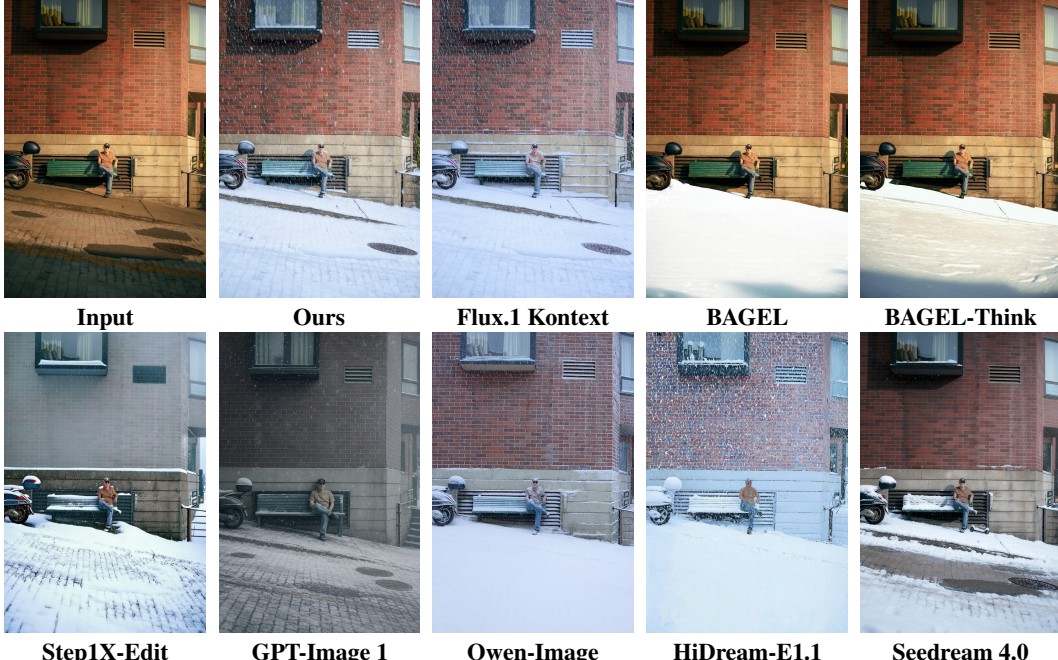

Figure 14: Examples of how models follow the law of global in state transition (superficial propmts).

**Superficial Prompt:** Make the bench wet.

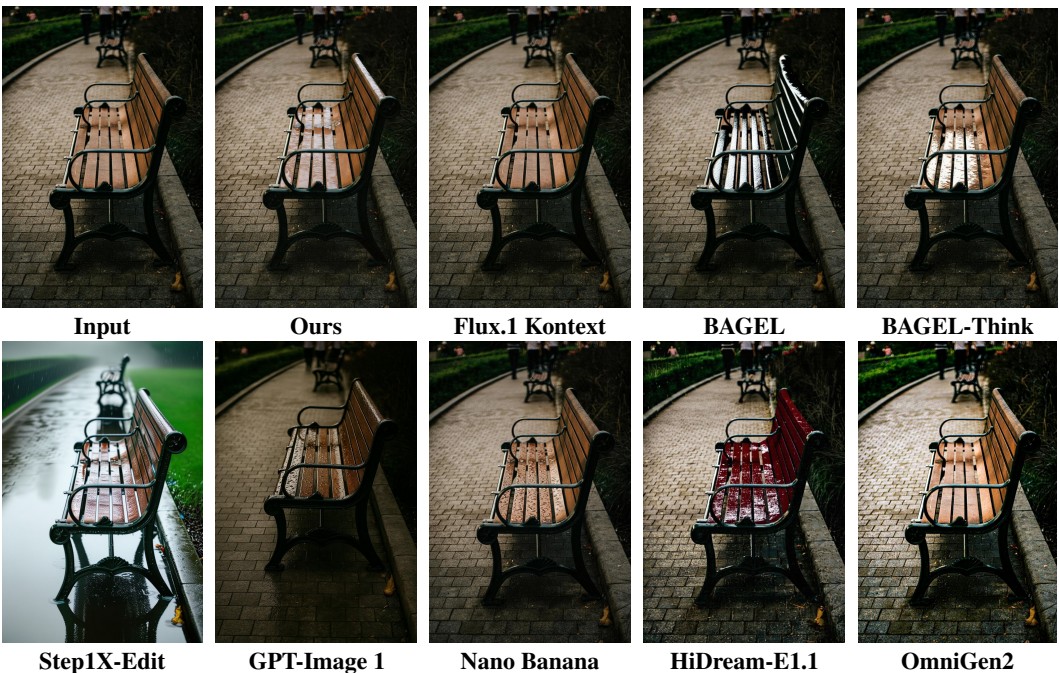

**Superficial Prompt:** Freeze the soda can.

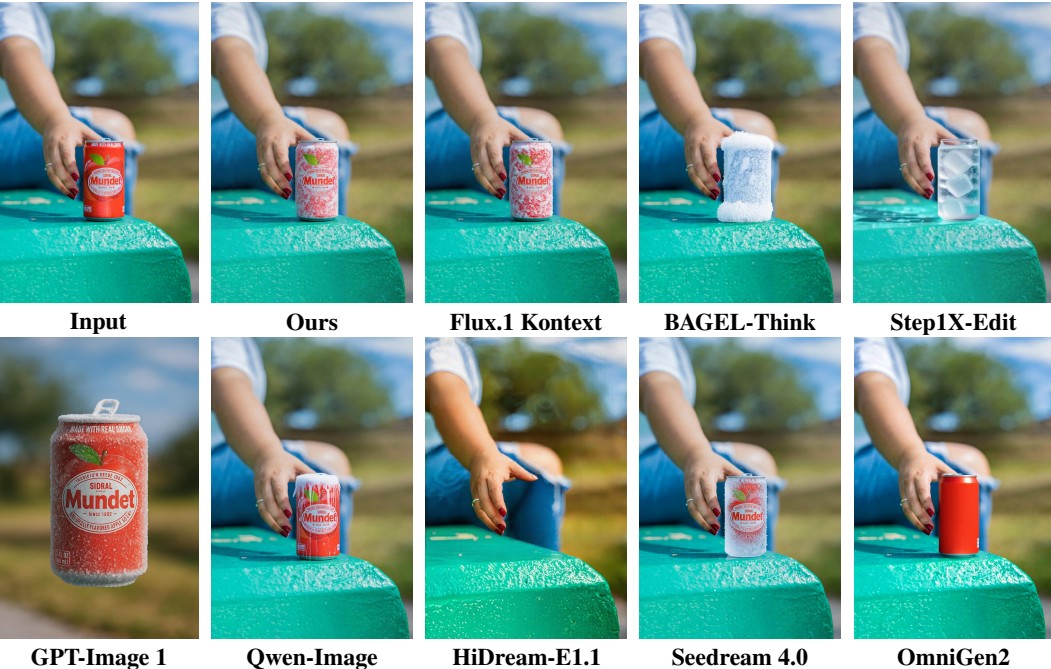

Figure 15: Examples of how models follow the law of local in state transition (superficial propmts).

**Prompt for PICABench Edit Instruction Generation**

Assume you are an experienced expert in image editing annotation. Your task is to generate three types of prompts — **explicit**, **intermediate**, and **superficial** — based on a given input prompt related to a {operation} operation.

You are provided with:
- **Image**: A real-world scene in which the editing will take place. You should use its visual content (objects, textures, layout, lighting, material, etc.) to guide your reasoning.
- **existing_instruction**: A user-provided high-level description of the intended image editing operation. It may be vague or encoded with physical or scientific implications.
- **physics_requirement**: A description of the physical laws or scientific constraints that the editing operation must respect. This ensures the output remains physically plausible and visually coherent.

Your goal is to interpret the instruction at **three levels of understanding**, each reflecting a distinct stage in physical reasoning and visual modeling.
**Important**: The output image should reflect the **physically stable result after the edit has taken full effect**, not the transitional moment when the edit has just occurred. For example, if a support is removed, falling or deformation should be shown as already completed. Avoid depicting unstable or transitional states.

---

**Superficial Prompt**:
Reformulate the instruction into a literal or surface-level command that includes only the basic editing action.

**Intermediate Prompt**:
Reformulate the instruction by adding awareness of the relevant physical laws or constraints, but **do not describe the specific visual or physical outcomes** that would result from these laws. The focus is on indicating that physical effects are expected, without predicting or rendering their actual consequences. This prompt bridges superficial and explicit by introducing real-world mechanism awareness without specifying the final appearance.

**Explicit Prompt**:
Reformulate the instruction into a detailed, physically consistent command that aligns with the expected **stable visual outcome** of the {operation} operation. Fully consider the image content, physics requirement, and any implied scientific or material properties (e.g., surface interaction, gravity-driven changes, deformation, light reflection). Explicit prompts should describe the **final, steady-state transformation** in a visually and physically coherent manner. Do not describe transient or instantaneous effects.

---

Here is the existing instruction: {existing_instruction}
Here is the physics requirement: {physics_requirement}

Please output in the following format:
{{"superficial_prompt": "your superficial prompt",
  "intermediate_prompt": "your intermediate prompt",
  "explicit_prompt": "your explicit prompt"}}

Figure 16: Prompt used to generate edit instruction for PICABench.

**Prompt for QA Generation**

You are an expert in image editing evaluation. Your task is to generate specific, targeted QA pairs to assess the success of this image editing task.

EDITING TASK CONTEXT:
- Edit Instruction: {edit_instruction}
- Physics Law: {physics_law}
- Operation Type: {operation}

CRITICAL CONSTRAINT: The evaluator will ONLY see the final edited image and the edit instruction. They CANNOT see the original image for comparison. Therefore, all questions must be answerable by looking at the edited image alone.

GENERATE QUESTIONS FOR TWO CATEGORIES:

1. EDITING COMPLETION ASSESSMENT:
Your goal: Verify that the specific changes requested in the instruction are visible in the final image.
- Always explicitly localize the target object referenced by the instruction using a locator phrase within the noun phrase (for disambiguation), not as an additional predicate.
- Build the locator phrase by combining any of the following as needed to uniquely identify the target: position (left/right/top/bottom/center/middle/foreground/background/upper-left quadrant), relative position (nearest to/left of/right of/in front of/behind/on top of/under/inside/next to [reference object]), ordinal (leftmost/rightmost/center-most/closest to center/first from the left), attributes (color/pattern/material/size/shape/text/logo), relationships (attached to/hanging from/placed on).
- Focus on observable characteristics in the edited result.

2. PHYSICS CONSISTENCY ASSESSMENT:
Your goal: Evaluate whether the final image respects the laws of {physics_law}.
- Check whether the current state of objects follows {physics_law} principles.
- Look for physically impossible or unrealistic arrangements.
- Assess whether object states, positions, orientations, contacts, shadows, reflections, and interactions are plausible.
- Focus on the current physical state, not the transition process.

MANDATORY SINGLE-CRITERION RULE:
- Each question MUST test exactly one observable predicate about the localized target.
- DO NOT combine multiple predicates using "and", "or", "both", "either", "while", or "except". If multiple checks are needed, split them into multiple questions.
- Connectors are allowed INSIDE the locator phrase only (for disambiguation), not in the predicate.
- Examples of single predicates: present/absent, is color X, located at Y, facing/oriented toward Z, touching/overlapping, casting a shadow, has reflection, number equals N.

QUESTION FORM GUIDELINES:
- For removal edits: ask absence of the localized target, e.g., "Is the [locator phrase] [object] absent?".
- For addition edits: ask presence of the localized target.
- For move/position edits: ask location with an absolute or relative anchor (e.g., "on the left side", "near the window", "in front of the sofa").
- For attribute edits (color/texture/material/text): ask for the new attribute on the localized target.
- For count edits: ask a single numeric predicate (e.g., "Are there exactly N [localized objects]?" ).
- Use concise, concrete language. Avoid vague terms like "some", "appears to", "looks like".

REQUIREMENTS:
- Be concise: Keep questions as brief as possible while maintaining clarity.
- Use simple language. You may split complex checks into multiple simple questions.
- Avoid ambiguity; ensure a single interpretation.
- Be specific and use a clear locator phrase to disambiguate instances when the category appears multiple times.
- Frame questions positively.
- Cover all key aspects of the instruction with multiple atomic questions as needed; each question should target a different aspect.

CRITICAL: Every answer must be exactly "Yes" or "No" — no other values are acceptable. Do not leave any answer empty.

OUTPUT FORMAT:
{{
  "Editing Completion QA": [
    {{"question": "...", "answer": "Yes"}},
    {{"question": "...", "answer": "No"}}
  ],
  "Physics Consistency QA": [
    {{"question": "...", "answer": "Yes"}},
    {{"question": "...", "answer": "No"}}
  ]
}}

EXAMPLES (DO NOT OUTPUT THIS SECTION):
- BAD (ambiguous target): "Is there a table?"
- GOOD (localized, absence predicate): "Is there a round wooden table in the center foreground?"
- BAD (two predicates): "Is the central table removed and is the floor clean?"
- GOOD (splitable, single predicate example): "Is there a  central round table?" (separate any floor-related check)
- GOOD (addition): "Is there a small blue cup on the right edge of the desk?"
- GOOD (move): "Is there a traffic cone placed on the left side of the crosswalk?"
- GOOD (attribute): "Is the leftmost of the two vases red?"
- GOOD (physics): "Is the shadow of the lamp cast toward the lower-right, consistent with a top-left light source?"

Remember: Questions should evaluate the final image state, not compare with the original or ask about the editing process. MOST IMPORTANTLY: Ensure each question evaluates a different aspect — avoid asking the same thing with different words.

Figure 17: Prompt used to generate QAs for PICABench

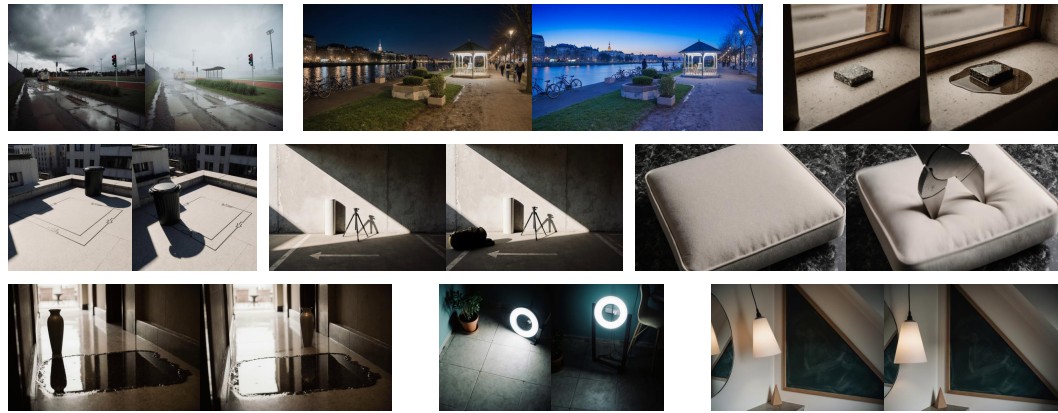

Figure 18: Examples of PICA-100K dataset.

**Prompt for I2V Caption Generation**

You are an expert writer of image-to-video captions (3–5 s).
You will receive ONE input image. DO NOT ask questions. DO NOT mention "image/photo/edit/prompt".

Goal
- Produce ONE concise motion caption that creates a VISUALLY OBVIOUS content change consistent with the physical law: "{law_cfg}".
- "Content change" means change the state of light source(add/remove/move/change color or intensity), add/remove/move/replace an object, or alter a local/global material state.
- The camera is secondary: keep camera static unless a tiny move is necessary for visibility.

Thinking steps (silently; no intermediate output)
1) Parse the scene: pick 1–2 salient, nameable objects; note supporting surface(s), visible light sources, mirrors/glass/water, soft/deformable materials, stacks/supports.
2) Choose ONE change that {law_cfg} can plausibly cause in THIS scene. Prefer the most visible option (area affected ≥ ~20% of subject OR global illumination clearly changes).
3) Make it measurable: include direction/magnitude cues (e.g., "slides right by half its width", "shadow doubles in length", "lamp turns off", "dense droplets form").
4) Keep identities and layout stable unless {law_cfg} implies a global change (only for "Global").
5) Camera: by default say "camera static". If using camera, use exactly ONE simple motion and keep content change primary.

Law playbook (pick ONE that fits the scene; prefer the first feasible, high-salience option)
- Light_Source_Effects: **turn on/off a visible lamp**; or change lamp color; or slightly move lamp so **all lit areas** update coherently.
- Light_Propagation: **add/remove/move/replace** an object on a flat surface so its **shadow position/length/shape** updates with a single key light.
- Reflection: **add/remove/move/replace** an object in front of a mirror or glossy surface; the **mirror image** updates/disappears consistently.
- Refraction: **add/remove/move/replace** an object or the transparent medium (glass/water) so the **region seen through** shows realistic **warping/dispersion**.
- Deformation: **add/remove weight** or apply pressure on a soft item (pillow, foam, fabric) to create a **deep indentation + partial rebound**.
- Causality: **remove a support / add off-center load** so a stack/board **tilts, slides, or collapses under gravity**.
- Local (state change): make local state change on target objects: include but not limited to **wet/dry/burn/frozen/splash/melt/fracture/wrinkle** with strong local visual cues (droplets, gloss, soot, frost, cracks).
- Global (scene-wide): **time/season/weather shift** so lighting, shadows, materials, vegetation, and atmosphere update **coherently** across the whole scene.

Hard constraints
- Duration 3–5 seconds, single continuous shot, stable exposure.
- **Primary change must be content change**, not camera-only motion.
- No new objects unless your chosen change requires "add"; no deletions unless your chosen change is "remove".
- Use the most concrete nouns visible in the scene (e.g., "desk lamp", "ceramic mug", "mirror", "glass of water", "pillow", "book stack").
- Avoid stories or meta language; do not name the law.

Style
- ONE sentence; explicit camera state at the end ("camera static" / "slow push-in" if truly needed).
- Include at least one physics cue word when relevant:
  shadows/highlights (light), reflection, warping/dispersion (refraction), indentation/rebound (deformation), tilt/slide/collapse (causality), droplets/frost/soot/cracks (state), color temperature/atmosphere (global).

Output format
Return ONLY valid JSON:
{{"i2v_prompt": "<one-sentence caption>"}}

Good outputs (examples; do NOT copy blindly)
- Light_Source_Effects: "The desk lamp turns off and all previously lit areas fall into dimness while shadows vanish or soften across the desk, camera static."
- Light_Propagation: "The ceramic mug slides right by about half its width and, under left key light, its sharp shadow shifts right and shortens, camera static."
- Reflection: "A red apple appears before the mirror and its mirror image pops into the correct mirrored position with matching highlights, camera static."
- Refraction: "The spoon moves left behind the glass of water and the portion seen through the glass warps and shifts with slight color dispersion, camera static."
- Deformation: "A dumbbell is placed on the pillow, forming a deep indentation with a clear contact patch and slow partial rebound after settling, camera static."
- Causality: "The small support block is removed from the book stack and the upper books tilt and slide down in a continuous gravity-driven collapse, camera static."
- Local·wet: "Dense droplets quickly form on the fabric and the surface darkens with bright specular glints while the surroundings remain unchanged, camera static."
- Global·time: "Light shifts to sunset; shadows grow longer and warmer uniformly across all objects, camera static."

Figure 19: Prompts used to generate i2v caption for Wan2.2 14B

**Prompt for PICA100K Edit Instruction Generation**

Assume you are an experienced expert in image editing annotation. Your task is to generate editing instruction(s) based on a given input-output image pair.

You are provided with:
- **Input Image**: The source image before editing
- **Output Image**: The target image after editing

Your goal is to analyze the visual changes between the input and output images and generate editing instruction(s) reflecting the specified depth of understanding about the transformation.

**Important**:
1. The instructions should describe HOW TO EDIT the input image to achieve the target result, not describe the images themselves.
2. Do NOT use phrases like "second image", "output image", "target image", "result image" in your instructions.
3. Write instructions as commands for editing the input image, focusing on what changes need to be made.
4. For explicit prompts: Focus on observable physical effects and realistic visual changes. Avoid technical implementation details, specific numerical values, or impossible operations like "generate depth maps" or "estimate transmittance maps".
5. Describe physical principles in terms of their visual manifestations, not technical processes.

---
**Superficial Prompt**: Keep the superficial prompt concise and under 20 tokens.
Generate a simple imperative sentence that describes the most obvious change needed. Use direct commands like "Add...", "Remove...", "Change...", "Make...". Focus only on the primary visual change.

**Intermediate Prompt**: Keep the intermediate prompt concise and under 128 tokens.
Generate an editing instruction that shows awareness of underlying processes or mechanisms, but without detailed physical explanations. This level bridges superficial and explicit by indicating that certain effects or processes are expected. Write as a command to edit the image.

**Explicit Prompt**: Keep the explicit prompt concise and under 256 tokens.
Generate a detailed editing instruction that describes the visual changes and the underlying physical principles that cause them. Focus on:
1. What visual changes occur (objects, lighting, materials, textures, composition)
2. The physical mechanisms behind these changes (light behavior, material properties, environmental effects, forces)
3. How these physical processes manifest as observable visual effects

Write as clear editing commands that demonstrate understanding of real-world physics. Avoid technical implementation details, specific numerical parameters, or impossible operations. Focus on describing the physical cause-and-effect relationships that produce the visual transformation.

**Examples of what to avoid:**
- Technical details: "generate depth maps", "apply Gaussian blur", "multiply by transmittance map"
- Specific numbers: "increase exposure by +0.4 EV", "reduce contrast by 30%"
- Implementation methods: "use Screen/Overlay mode", "apply particle layers"

**Examples of what to include:**
- Physical effects: "heavy snowfall reduces visibility through light scattering"
- Visual manifestations: "distant objects become progressively fainter and less defined"
- Cause-and-effect: "atmospheric particles scatter light, creating a hazy, washed-out appearance"

---

Please analyze the input-output pair and generate the three levels of editing instructions.

Please output in the following format:
{{"superficial_prompt": "your superficial prompt",
  "intermediate_prompt": "your intermediate prompt",
  "explicit_prompt": "your explicit prompt"}}
"""

Figure 20: Prompts used to generate edit instruction for PICA100K

