# OpenReview forum: "PICABench: How Far are We from Physical Realistic  Image Editing?"
_ICLR.cc/2026/Conference — ICLR 2026 Poster_

### Official Review · Reviewer_QMZg · 2025-10-28

**Soundness:** 3
**Presentation:** 2
**Contribution:** 2
**Rating:** 4
**Confidence:** 3

**Summary:**

This paper addresses a critical gap in image editing by introducing PICABench, a benchmark designed to evaluate the physical realism of edited images, moving beyond mere semantic fidelity. The authors argue that while modern models can follow complex editing instructions, they largely fail to simulate accompanying physical effects (e.g., shadow/reflection updates, mechanical interactions), which is crucial for realism. To systematically assess this, PICABench categorizes physical consistency into eight fine-grained sub-dimensions spanning Optics, Mechanics, and State Transitions, covering most common editing operations. The paper also proposes PICAEval, a reliable evaluation protocol that leverages a VLM-as-a-judge with per-case, region-specific questions grounded by human annotations, thereby reducing hallucinations and improving sensitivity to nuanced physical violations. Beyond benchmarking, the authors explore a solution by learning physics from videos, constructing the PICA-100K dataset. Their experiments show that fine-tuning on PICA-100K significantly enhances a model's physical consistency. The comprehensive evaluation of 11 models reveals that physical realism remains a significant challenge, establishing this work as a foundational step towards physically plausible image editing.

**Strengths:**

- The research on complex instruction-based image editing is highly meaningful, and the proposed method demonstrates improvements over current state-of-the-art approaches.
- The paper presents substantial work, including the introduction of a new benchmark, the provision of corresponding training data, and fine-tuning of the FLUX model.
- The writing is clear, and the experimental comparisons and figure illustrations are professional.

**Weaknesses:**

- The PICA-100K construction pipeline relies on video generation models, meaning the final benchmark performance heavily depends on the quality of WAN. I am concerned about whether such data can ensure sufficient effectiveness, even if it performs well on the proposed benchmark. Given the proliferation of benchmarks for world-knowledge image editing, I question whether this data would generalize well to other benchmarks.
- The authors fine-tuned the FLUX image generation model and achieved promising results. However, I believe that tasks involving world-knowledge image editing should be improved based on VLM+Diffusion model frameworks, such as Bagel, UniWorld, or Qwen-Image-Edit. While the proposed method outperforms these unified generation-understanding models on PICABench, it may be attributed to the PICA-100K dataset being tailored specifically for this benchmark, allowing a pure diffusion model to overfit to it. It may be unreasonable to request extensive additional experiments during the rebuttal, so I encourage the authors to focus on analyzing this point. My final score will depend on the analysis provided.
- There has been a surge of related work on world-knowledge editing in recent years. While citing the latest work is not mandatory, some early relevant studies should be referenced:

[1] Echo-4o: Harnessing the power of gpt-4o synthetic images for improved image generation \
[2] Anyedit: Mastering unified high-quality image editing for any idea \
[3] Editworld: Simulating world dynamics for instruction-following image editing \
[4] Reasonpix2pix: instruction reasoning dataset for advanced image editing \
[5] Smartedit: Exploring complex instruction-based image editing with multimodal large language models

**Questions:**

See the "Weaknesses" section. Additionally, as I have reviewed other higher-quality papers on world knowledge editing and find the choice of the fine-tuned model in this paper somewhat inappropriate, my current score is relatively low. I will adjust my rating based on the rebuttal and other reviewers' comments.

---

> ### Author Response · Authors · 2025-11-22
>
> We sincerely thank the reviewer for the constructive feedback and for recognizing the significance of our benchmark and the professional quality of our experiments. We take the reviewer's concerns regarding the choice of base models and the data generalization very seriously.
>
> > Q1 Reliance on video generator.
>
> We acknowledge that synthetic data is bounded by the generator's quality. However, our pipeline is designed to mitigate this and outperform real-world data extracted from real-world data.
>
> Recent advances have demonstrated that large-scale video generation models are emerging as world simulators, possessing a stronger understanding of physical laws and dynamics than static image models. Our pipeline treats the video model [4] as a "teacher" to distill these internalized physical priors into the image editing model. By using prompt-controlled generation, we can isolate specific physical effects much more cleanly than filtering through terabytes of real video.
>
> We use the first/last-frame strategy. While video models may hallucinate temporal inconsistencies in intermediate frames, the start and end states generated by WAN2.2-14B are generally physically coherent and high-quality, making them suitable for training image editing models.
>
> While real-world videos capture true dynamics, they are inherently noisy (containing irrelevant motions, occlusions, and complex lighting changes) and difficult to collect for specific physical phenomena. As shown in Tab.3 of our paper, the model trained on PICA-100K significantly outperforms the model trained on Mira400K (Real Video Data). This demonstrates the efficiency and effectiveness our data pipeline.
> > Q2 Generalization on other benchmarks
>
> We test Qwen-Image-Edit model and our finetuned model on Kris-Bench[6]. The result is shown below. As shown in table below, fine-tuning on PICA-100K indeed improves the model’s capability on instruction following, which means model can successfully perform editing operations.
>
> VC = Visual Consistency, VQ = Visual Quality, IF = Instruction Following.
>
> | Model | Attr. Perception |  |  | Spatial Perc. |  |  | Temporal Pred. |  |  | Social Sci. |  |  | Natural Sci. |  |  | Logical Reason. |  |  | Instr. Decomp. |  |  | Overall |  |  |
> |-------|:---:|:---:|:---:|:---:|:---:|:---:|:---:|:---:|:---:|:---:|:---:|:---:|:---:|:---:|:---:|:---:|:---:|:---:|:---:|:---:|:---:|:---:|:---:|:---:|
> |       | VC | VQ | IF | VC | VQ | IF | VC | VQ | IF | VC | VQ | IF | VC | VQ | IF | VC | VQ | IF | VC | VQ | IF | VC | VQ | IF |
> | **Qwen-Image-Edit** | 79.6 | 90.2 | 64.2 | 78.2 | 94.4 | 56.4 | 49.0 | 87.0 | 47.0 | 73.6 | 92.6 | 60.8 | 77.8 | 92.6 | 58.6 | 83.4 | 90.8 | 45.0 | 84.0 | 90.4 | 95.0 | 77.0 | 91.4 | 60.0 |
> | **Qwen-Image-Edit + SFT** | 81.4 | 89.4 | 68.2 | 78.6 | 91.0 | 63.8 | 46.2 | 83.6 | 49.4 | 74.8 | 88.8 | 63.8 | 76.2 | 88.8 | 60.4 | 83.4 | 90.6 | 48.4 | 83.2 | 88.0 | 93.8 | 76.8 | 89.0 | 63.0 |
> | **Δ Improvement** | +1.8 | -0.8 | +4.0 | +0.4 | -3.4 | +7.4 | -2.6 | -3.2 | +2.4 | +1.2 | -3.8 | +3.0 | -1.6 | -3.8 | +1.8 | +0.0 | -0.2 | +3.4 | -0.8 | -2.4 | -1.2 | -0.2 | -2.4 | +3.0 |
>
> [6]Wu Y, Li Z, Hu X, et al. KRIS-Bench: Benchmarking Next-Level Intelligent Image Editing Models[J]. arXiv preprint arXiv:2505.16707, 2025.
>
> > Q3 Choice of base model
>
> We fully agree with the reviewer that Unified VLM+Diffusion models represent a critical direction for world-knowledge editing. Our initial choice of Flux was to establish a clean baseline on a widely used image editing model.
>
> To explicitly address the reviewer's concern and verify that PICA-100K is not overfitting to a pure diffusion model, we fine-tuned Qwen-Image-Edit on PICA-100K using the same settings. Please refer to responses to Reviewer Ytia. These results demonstrate that the physics priors distilled by PICA-100K are not tied to a particular architecture: PICA-100K benefits both a pure diffusion model (FLUX.1 Kontext) and strong VLM+Diffusion model.
>
> We emphasize that training on PICA-100K is not overfitting the benchmark. First, we use different data pipelines to curate PICABench and PICA-100K. PICABench is collected from real world scenarios. Data in PICA-100K is generated from video generation models. Data distributions are different. Second, although PICA-100K is aligned with the same sub-dimensions, it covers a much larger variety of scenes and objects than PICABench. The model must learn general pattern, not memorize specific layouts. Third, fine-tuning on PICA-100K doesn’t show a huge performance gain on PICABench. This also suggests that PICA-100K is not overfitting the benchmark.
>
> > Q4: Related work
>
> We appreciate the reviewer’s suggestion. We will expand related work in the revised version.

---

> > ### Author Response · Authors · 2025-11-27
> >
> > Dear Reviewer
> >
> > Thank you again for your valuable comments during the discussion.  With only a few days left before the discussion deadline, we kindly wanted to ensure we have addressed all your concerns satisfactorily. We respectfully ask whether there are any additional concerns or suggestions you would like us to address. We greatly value your insights and are eager to make further improvements if needed.
> >
> > We appreciate your time and dedication.

---

> > > ### Comment · Reviewer_QMZg · 2025-11-27
> > > **Response to Authors**
> > >
> > > We appreciate the authors' response. The revision, which involved fine-tuning Qwen-Image, has satisfactorily addressed our primary concern regarding this issue. Consequently, we have increased our overall score for this submission. We further encourage the authors to continue their efforts in optimizing the PICA-100K dataset.

---

> > > > ### Author Response · Authors · 2025-11-28
> > > >
> > > > Thank you very much for your encouraging feedback and for raising the score.
> > > > We are grateful for your insights and will continue to refine our work based on your suggestions.

---

### Official Review · Reviewer_ghns · 2025-11-01

**Soundness:** 3
**Presentation:** 3
**Contribution:** 3
**Rating:** 4
**Confidence:** 3

**Summary:**

This paper introduces PICABench, a benchmark for systematically evaluating the physical realism of instruction-based image editing models. The authors argue that existing models and benchmarks primarily focus on semantic fidelity and instruction completion, neglecting crucial physical effects like correct shadows, reflections, and object interactions, which are key to true realism. To enable reliable evaluation, the paper proposes PICAEval, a region-grounded, VLM-as-a-judge protocol that uses per-case, region-level human annotations and binary Yes/No questions (VQA) aligned with specific physical sub-dimensions. Finally, authors propose the PICA-100K, a large-scale, synthetic training dataset (100k examples) and experiments show that fine-tuning an existing model (FLUX.1 Kontext) on PICA-100K significantly enhances its physical consistency.

**Strengths:**

1. PICABench covers eight **fine-grained sub-dimensions** and a variety of common editing operations. And PICAEval addresses the known limitations (hallucination, low sensitivity) of standard VLM-as-a-judge setups by incorporating region-level annotations and targeted Q&A.

2. The paper is very clear and **well-structured**. The core dimensions (Optics, Mechanics, State Transition) are immediately intuitive. Key concepts like the eight sub-dimensions are defined with concrete, checkable criteria

3. It introduces a critical **new evaluation standard** that forces the community to move beyond mere content editing toward physically consistent realism.

**Weaknesses:**

1. The authors note a slight drop in the State Transition Accuracy for their fine-tuned model and speculate this is due to only using the first and last frames of a video to represent meaningful state changes. This highlights a potential weakness in the PICA-100K data generation pipeline for this specific dimension. The use of only two frames might fail to capture the coherence and physical consistency of the transition process itself, as the model only sees the 'before' and 'after' states. While they mention plans to explore fine-grained strategies, the current limitation is a weakness in the proposed training data solution.

2. The metrics used (Chamfer Distance, F-Score) are standard but insufficient to evaluate perceptual quality or shape plausibility. No perceptual or task-level metric is proposed

3. The paper notes that unified multi-modal models consistently underperform compared to dedicated image editing models, speculating that their enhanced world understanding doesn't translate to physical realism due to a lack of "internalized physics principles". This is a crucial observation, but the analysis remains largely speculative. A more in-depth qualitative analysis, perhaps showing how a unified model (like Bagel or OmniGen2) fails to integrate its supposed world knowledge into the generation process (e.g., how its explicit reasoning about a scene's physics is decoupled from its final generative output), would strengthen this claim

4. Table 2 shows that performance improves with the specificity of the prompt, with a notably small gain between "superficial" and "intermediate" prompts compared to "explicit" prompts. While the authors speculate this is due to a lack of internalized physics, it could also indicate that the models are highly sensitive to explicit guidance (i.e., the "explicit" prompts describe the desired physical outcome) rather than being able to infer the physics from the instruction and scene context (which the "intermediate" prompts attempted to provide). This suggests a potential weakness in the model's ability to generalize physical principles from scene context alone

**Questions:**

Please see the weakness

---

> ### Author Response · Authors · 2025-11-22
>
> We sincerely thank the reviewer for the detailed and constructive feedback. We are encouraged that the reviewer finds our benchmark "fine-grained" and "well-structured," and recognizes the "new evaluation standard" we introduced. We also appreciate the insightful comments regarding the behavior of unified models and prompt sensitivity. Below, we address the specific concerns, **particularly to clarify a misunderstanding regarding the evaluation metrics**.
>
>
>
> > Q1 First/last-frame strategy.
>
> We acknowledge this limitation. As stated in Sec. 3.4, previous works have proven that learning from videos can help models achieve better editing performance. The motivation of PICA-100K is to verify if models can learn physics from videos. Our results have proven this hypothesis.
>
> The choice of the first/last-frame strategy is to balance scalability and robustness. Current video generators may exhibit hallucinations in intermediate frames; naively supervising on all frames risks teaching these artifacts to the editor. Focusing on the initial and final frames mitigates this issue while still capturing the key physical changes.
>
> We will continue exploring a better strategy to mitigate limitations in the current data pipeline.
>
> > Q2 Clarification on Evaluation Metrics:
>
> We respectfully point out a possible misunderstanding regarding the metrics used in our paper. We do not use Chamfer Distance or F-Score in our evaluation. As detailed in Section 3.3 and Section 4.1, our primary metric is PICAEval (Acc), a VLM-as-a-judge metric. It utilizes state-of-the-art VLMs to answer region-grounded, physics-specific questions. We also use PSNR (Cons) solely to measure background consistency.
>
> > Q3 Qualitative analysis of unified model
>
> We thank the reviewer’s suggestion. We provide Bagel’s results in thinking mode in our revised version, where model first reasons about editing instruction and then performs editing operation. As shown in Fig. 7, Bagel successfully infers correct results, but fails in generating them. This indicates that it fails to integrate its supposed world knowledge into the generation process.
>
> > Q4 Potential weakness of generalizing physical principles.
>
> We fully agree with the reviewer’s interpretation, which powerfully reinforces our core motivation.
>
> If the models had robust, internalized physical priors, we would expect substantial performance gains on intermediate prompts. However, as shown in Tab.2, the fact that i*ntermediate* prompts yield minimal gains compared to explicit prompts confirms that current models are instruction followers and do not understand physical principles.

---

> ### Author Response · Authors · 2025-11-27
>
> Dear Reviewer,
>
> As the discussion period will conclude soon, we just wanted to check in briefly. Please let us know if there are any remaining issues you would like us to respond to. We appreciate your time and consideration.
>
> Thank you again for reviewing our submission.

---

### Official Review · Reviewer_guET · 2025-11-01

**Soundness:** 3
**Presentation:** 3
**Contribution:** 3
**Rating:** 6
**Confidence:** 3

**Summary:**

This paper focuses on the issue of low physical consistency in instruction-based 2D editing results, and proposes PICABench, a benchmark for physical realism preservation capabilities of image editing models. By analyzing common cases of low realism, the paper assesses the physical consistency in terms of optical, mechanical, and state transition aspects of edited image content. Specifically, this paper proposes the VLM-based evaluation metric PICAEval, combining human annotation mechanisms and QA patterns to evaluate the physical realism of 2D editing results. Furthermore, this paper constructs the synthetic image editing dataset PICA-100K based on video generation modalities to fine-tune and improve the physical preservation capabilities of large-scale 2D editing models.
Through the evaluation of existing models, this paper shows that "most existing large-scale models are still unable to achieve physically realistic 2D editing" and analyzes the gap between instruction understanding and physical preservation in existing models. Moreover, by fine-tuning models with the proposed dataset PICA-100K, this paper demonstrates the effectiveness of PICA-100K.

**Strengths:**

- This paper is a benchmark paper. It analyzes common physical realism issues in existing 2D editing models, and proposes PICABench and PICAEval to evaluate model capabilities from multiple dimensions. The motivation is clear, and it is important for image editing models.

- The paper explores the capabilities of VLM models, the QA evaluation method of LLM, the human annotation process and the advantages of video for dataset construction and method evaluation, which is novel and effective.

- The paper is well-organized and easy to follow.

**Weaknesses:**

1.  Physical awareness is important for 2D editing mode, but I still concern about requirements that editing with physical awareness directly.
- It is more like the issue of prompts/instructions. In addition, the definition of hallucination in this paper should be clearer. The 2D editing model needs to balance between precisely edition according to the prompts, and the certain ability to extrapolate. For example, in Fig.1 bottom right (remove the scooter), it is kind of hallucination if the rider stands on the floor, which also violates our prompt. Maybe, the user only simulates a snapshot that the rider flies.
- The middle or long instructions in Fig.4 (d) are more reasonable. Tab.2 should include more methods in the main paper.
- Tab.2 constructs different levels of prompt, but the performance is highly impacted by the long-text comprehension. It feels like the problem has changed to 2D editing with long prompts. Therefore, it would be better to compare with methods based on multiple steps instead of one step with long prompts.

2. There is a lack of detailed explanation of metrics Acc and Con.

3. Line 424 claims that "model performance improves as prompts become more detailed". However, Tab.2 shows that the consistency (Con) decreases in most cases. Moreover, Line 425 claims that "the gain from intermediate prompts is much smaller than that from explicit prompts". However, it is hard to verify on the basis of tables.

**Questions:**

See weaknesses

---

> ### Author Response · Authors · 2025-11-22
>
> We greatly appreciate  the reviewer’s recognition of our work, especially the positive comments on our presentation, novelty,  and method effectiveness.
>
> > Q1: The definition of hallucination.
>
> We thank the reviewer’s insightful question. We provide the definition of the tasks and the corresponding failure case in Sec. A.1. In particular, causality requires that edited objects have plausible contacts and supports under gravity; objects should not float, interpenetrate, or rest in unstable equilibria. Under this definition, we label hallucinations when an edited state clearly lacks physical support, even if the semantic instruction is roughly satisfied.
>
> For the bottom-right case in Fig. 1, the current model output makes the rider appears to hover in an upright pose without visible support or motion cues, which we consider a violation of the causality rule above (unsupported, unstable equilibrium). If the edited result instead showed a plausible airborne snapshot (e.g., a jumping or mid-air pose with appropriate body configuration and trajectory cues), it would not be penalized; our human annotations and PICAEval questions are written precisely to distinguish between “physically possible airborne motion” and “unsupported hovering.”
>
> We will clarify this point and use this example to illustrate our notion of physical hallucination.
>
> > Q2: Prompt length.
>
> We agree that intermediate /explicit instructions are more reasonable to instruct models. This is exactly why PICABench is designed with three prompt specificity levels.
>
> Superficial prompts use pure edit instruction without explanation, reflecting models’ *intrinsic* physical priors. Intermediate and explicit prompts probe whether models can effectively exploit additional physical hints. As shown in Tab. 2, there is a significant gap between superficial prompts and explicit prompts. This indicates that models fail to internalize physics principles.
>
> Furthermore, in real-world scenarios, users usually provide short prompts to editing models. A physics-aware image editing model should infer the necessary physical consequences from the scene context, rather than relying on the user to explicitly describe the visual outcome.
>
> We will clarify our motivation of different prompt levels in our revised version.
>
> > Q3: Comparison with multi-step method and extension of Tab. 2.
>
> We provide results with Uni-CoT[5] as follows.
>
> | Model               | Prompt Level | LP Acc | LP Con | LSE Acc | LSE Con | Reflection Acc | Reflection Con | Refraction Acc | Refraction Con | Deformation Acc | Deformation Con | Causality Acc | Causality Con | GST Acc | GST Con | LST Acc | LST Con | Overall Acc | Overall Con |
> | ------------------- | ------------ | :----: | :----: | :-----: | :-----: | :------------: | :------------: | :------------: | :------------: | :-------------: | :-------------: | :-----------: | :-----------: | :-----: | :-----: | :-----: | :-----: | :---------: | :---------: |
> | **Bagel**           | explicit     | 58.03  | 17.63  |  66.90  |  18.44  |     57.90      |     20.95      |     51.40      |     18.68      |      58.29      |      18.81      |     53.37     |     23.71     |  66.12  |  33.53  |  57.00  |  21.41  |    59.56    |    23.06    |
> | **Flux.1 Kontext**  | explicit     | 57.64  | 27.77  |  66.90  |  25.25  |     59.67      |     27.35      |     40.21      |     26.96      |      56.51      |      28.67      |     57.74     |     27.99     |  65.68  |  36.11  |  52.24  |  29.57  |    59.11    |    29.40    |
> | **Qwen-Image-Edit** | explicit     | 57.06  | 20.18  |  69.72  |  22.79  |     63.09      |     22.63      |     52.10      |     26.10      |      58.11      |      24.95      |     59.29     |     23.99     |  66.79  |  35.03  |  60.46  |  25.08  |    62.09    |    25.78    |
> | **Nano Banana**     | explicit     | 59.00  | 29.01  |  62.15  |  30.04  |     61.67      |     27.43      |     51.40      |     27.97      |      62.57      |      28.52      |     63.45     |     29.14     |  61.14  |  38.66  |  56.57  |  31.02  |    60.62    |    30.90    |
> | **Uni-cot**         | superficial  | 54.56  | 24.83  |  59.82  |  21.77  |     49.01      |     30.36      |     55.12      |     18.30      |      49.12      |      19.34      |     48.79     |     20.20     |  60.98  |  11.84  |  51.70  |  17.28  |    53.56    |    20.31    |
>
> As shown above, Uni-cot underperforms the explicit prompt baselines on PICABench. This indicates that multi-step reasoning is not enough to achieve physical realism. We still need to internalize physical principles into editing models.
>
> Furthermore, we extend Tab.2 with more result in revised version.
>
> [5]Qin L, Gong J, Sun Y, et al. Uni-cot: Towards unified chain-of-thought reasoning across text and vision[J]. arXiv preprint arXiv:2508.05606, 2025.

---

> > ### Author Response · Authors · 2025-11-22
> >
> > > Q4: Detailed explanation of metrics.
> >
> > Thanks for the reviewer’s suggestion, we provide details of the metrics in the revised version. Please refer to Sec. A.3 for more details.
> >
> > > Q5: Trade off between accuracy and consistency.
> >
> > We thank the reviewer for pointing out this ambiguity. In our statement, “model performance” refers to accuracy on PICABench rather than consistency. We will refine the text in our revised version.
> >
> > There is a trade-off between physical realism and preservation. As prompts become more detailed, models are encouraged to adjust more aspects of the scene, which naturally modifies a larger portion of the image outside the minimal edited region. This often lowers PSNR in non-edited areas, even when the physics of the edited region is improved. We will add a short discussion of this trade-off and soften the claim accordingly.

---

> > > ### Author Response · Authors · 2025-11-27
> > >
> > > Dear Reviewer,
> > >
> > > We hope you are doing well. As the discussion period is approaching its end, we wanted to kindly check whether you have any additional questions or comments you would like us to address. We are happy to clarify anything that might help in your evaluation.
> > >
> > > Thank you very much for your time and thoughtful review.

---

### Official Review · Reviewer_Ytia · 2025-11-01

**Soundness:** 3
**Presentation:** 3
**Contribution:** 3
**Rating:** 6
**Confidence:** 4

**Summary:**

This manuscript proposes a benchmark to evaluate physical realism in image editing. The authors consider physical effects beyond simple instruction completion. They have collected a physics-aware benchmark, PICABench, to evaluate the performance of existing image editing models. Meanwhile, the authors also utilize a VLM-as-a-judge (PICAEval) to evaluate the edited images. Beyond the benchmark, the authors also propose constructing a physics-aware dataset (PICA-100K) from videos. Experimental results demonstrate that existing models still struggle to achieve physical realism. Furthermore, the proposed dataset is shown to enhance the physical realism of a fine-tuned base model.

**Strengths:**

This manuscript considers the role of physics in the image editing task, which is an interesting and rarely explored area of research. The proposed benchmark is well-constructed and provides a comprehensive analysis of existing models regarding the physical realism of their edits. In addition, the proposed PICA-100K dataset is a clever approach to learn physics from synthetic videos.

**Weaknesses:**

1. The primary weakness is that the proposed solution, fine-tuning on PICA-100K, yields very small gains. While this is a positive result and better than the baseline model, the small margin doesn't present PICA-100K as a definitive solution. Moreover, the final fine-tuned model still underperforms other top-tier models (e.g., GPT-Image-1, Seedream 4.0, Qwen-Image-Edit) in the overall benchmark, as shown in Table 1.

2. To better demonstrate the dataset's effectiveness, the authors should try fine-tuning other base models (besides Flux.1 Kontext) and report their performance improvements.

3. Meanwhile, the concept of constructing image editing pairs from synthetic videos is not entirely novel and has been explored in previous works (e.g., Frame2Frame[1], ByteMorph[2]). The reviewer recommends the authors explicitly to describe the differences and contributions of their data generation pipeline compared to these prior works.

[1] Pathways on the Image Manifold: Image Editing via Video Generation

[2] ByteMorph: Benchmarking Instruction-Guided Image Editing with Non-Rigid Motions

**Questions:**

See the weakness.

---

> ### Author Response · Authors · 2025-11-22
>
> We sincerely thank the reviewer for acknowledging the strengths of our work, including the research motivation, the comprehensive evaluation and analysis, and the novelty of our data construction pipeline.
>
> >Q1: Small performance gains.
>
> We appreciate the reviewer’s comments. We would like to clarify the primary motivation and contributions of our work as follows.
>
> First, our core contribution is to evaluate existing image editing models through the lens of physical realism. As stated in Sec. 4.2, all open-source models score below 60, even though they are trained on massive datasets. This indicates that we are still far from physically realistic image editing.
>
> Second, our goal with PICA-100K is not to surpass SOTA models but to answer the question “Can image editing models learn physics from synthetic video data?” As shown in Tab. 1, fine-tuning on PICA-100K yields consistent improvements in physical realism over the baseline. Furthermore, we show that our model performs better than a model trained on a much larger real-world video dataset (Mira400K). This demonstrates efficiency and effectiveness of our data pipeline.
>
> We hope that our benchmark and proposed solutions can serve as a foundation for future work, encouraging future research to move beyond semantic consistency to physical realism.
>
> >Q2:Fine-tuning other base models
>
> We sincerely thank the reviewer for this valuable suggestion. To demonstrate the generalizability and effectiveness of our dataset, we have conducted additional experiments on Qwen-Image-Edit. As suggested by Reviewer guET, we present results at different prompt levels to better understand the model’s behavior.
>
> | Model               | Prompt Level | LP Acc | LP Con | LSE Acc | LSE Con | Refl Acc | Refl Con | Refr Acc | Refr Con | Def Acc | Def Con | Cau Acc | Cau Con | GST Acc | GST Con | LST Acc | LST Con | Overall Acc | Overall Con |
> | ------------------- | ------------ | ------ | ------ | ------- | ------- | -------- | -------- | -------- | -------- | ------- | ------- | ------- | ------- | ------- | ------- | ------- | ------- | ----------- | ----------- |
> | Qwen-Image-Edit     | superficial  | 58.99  | 22.61  | 64.08   | 27.43   | 62.62    | 23.86    | 60.84    | 25.06    | 54.37   | 24.65   | 48.18   | 26.79   | 62.85   | 35.42   | 52.53   | 25.75   | 57.99       | 27.24       |
> | Qwen-Image-Edit+SFT | superficial  | 64.70  | 25.71  | 70.12   | 25.67   | 66.47    | 27.61    | 52.10    | 31.03    | 59.23   | 27.13   | 51.47   | 29.59   | 60.69   | 31.65   | 55.36   | 30.45   | 60.04       | 28.83       |
> | Improvement         | superficial  | +5.71  | +3.10  | +6.04   | -1.76   | +3.85    | +3.75    | -8.74    | +5.97    | +4.86   | +2.48   | +3.29   | +2.80   | -2.16   | -3.77   | +2.83   | +4.70   | +2.05       | +1.59       |
> | Qwen-Image-Edit     | Intermediate | 60.93  | 23.62  | 67.78   | 24.82   | 60.50    | 24.82    | 46.50    | 27.22    | 57.58   | 27.49   | 51.30   | 26.66   | 60.55   | 36.20   | 50.51   | 28.24   | 57.56       | 28.06       |
> | Qwen-Image-Edit+SFT | Intermediate | 64.46  | 24.03  | 69.06   | 23.18   | 66.08    | 27.28    | 46.64    | 30.62    | 59.57   | 27.71   | 57.03   | 27.82   | 60.40   | 32.11   | 57.95   | 31.47   | 60.85       | 28.29       |
> | Improvement         | Intermediate | +3.53  | +0.41  | +1.28   | -1.64   | +5.58    | +2.46    | +0.14    | +3.40    | +1.99   | +0.22   | +5.73   | +1.16   | -0.15   | -4.09   | +7.44   | +3.23   | +3.29       | +0.23       |
> | Qwen-Image-Edit     | Explicit     | 57.06  | 20.18  | 69.72   | 22.79   | 63.09    | 22.63    | 52.10    | 26.10    | 58.11   | 24.95   | 59.29   | 23.99   | 66.79   | 35.03   | 60.46   | 25.08   | 62.09       | 25.78       |
> | Qwen-Image-Edit+SFT | Explicit     | 63.03  | 22.54  | 74.89   | 21.06   | 66.31    | 24.56    | 43.40    | 26.60    | 65.01   | 24.01   | 64.34   | 24.30   | 67.99   | 31.22   | 64.87   | 28.84   | 65.30       | 25.81       |
> | Improvement         | Explicit     | +5.97  | +2.36  | +5.17   | -1.73   | +3.22    | +1.93    | -8.70    | +0.50    | +6.90   | -0.94   | +5.05   | +0.31   | +1.20   | -3.81   | +4.41   | +3.76   | +3.21       | +0.03       |
>
> Our results confirm that our dataset are not specific to a single architecture but are universally beneficial for enhancing the physical knowledge of instruction-based image editing models.

---

> ### Author Response · Authors · 2025-11-22
>
> > Q3 Comparison with other works
>
> We thank the reviewer for this insightful comment. We will expand the Sec. 3.4 to discuss the main distinctions from these prior works. The key differences are as follows:
>
> - Divergent Motivations and Contributions:
>
>   - While ByteMorph [2] focuses primarily on **non-rigid image editing**, emphasizing explicit and visually salient motions such as deformations, pose changes, and viewpoint shifts, our work instead targets **physical realism**, which represents implicit physics principles of the real world. Similarly, Frame2Frame [1] focuses on zero-shot Feasibility. It explores how to use video generation models to perform image editing without training.
>
> - Distinct Methodological Paradigms.
>
>   - Frame2Frame [1] proposes an inference-time, training-free solution. It directly leverages a video generation model to simulate the transition between the source and the edited images during the editing process itself.
>   - We employ video generation models in an offline data construction phase. We use video generators solely to synthesize a dataset to fine-tune image editing models. Our goal is to make image editing models learn physics from video data, enabling them to achieve physical realism.
>
> - Differences in Data Pipeline
>
>   - ByteMorph[2] still uses real images to generate videos. However, PICA-100K adopts a **fully synthetic** data pipeline. As detailed in Sec3.5 and Fig.5, we first generate diverse source images using Flux.1-Krea[3] and then use Wan2.2-14B[4] to synthesize video data. We have proven its efficiency and effectiveness in Tab. 1 and Tab. 3.
>   - Furthermore, ByteMorph ignores background consistency due to its focus on non-rigid image editing. Data in ByteMorph-6M shows large background changes. This may hurt the model’s ability to keep non editing regions unchanged. Our synthetic pipeline allows for controlled generation where the non-edited regions remain stable. As shown in Tab. 1, fine-tuning on PICA-100K improves the model’s consistency from 31.90 to 32.71.
>
>   [1]Rotstein N, Yona G, Silver D, et al. Pathways on the image manifold: Image editing via video generation[C]//Proceedings of the Computer Vision and Pattern Recognition Conference. 2025: 7857-7866.
>
>   [2]Chang D, Cao M, Shi Y, et al. ByteMorph: Benchmarking Instruction-Guided Image Editing with Non-Rigid Motions[J]. arXiv preprint arXiv:2506.03107, 2025.
>
>   [3] Stephen Batifol, Andreas Blattmann, Frederic Boesel, Saksham Consul, Cyril Diagne, Tim Dockhorn, Jack English, Zion English, Patrick Esser, Sumith Kulal, et al. Flux. 1 kontext: Flow matching for in-context image generation and editing in latent space. arXiv e-prints, 2025.
>
>   [4]Team Wan, Ang Wang, Baole Ai, Bin Wen, Chaojie Mao, Chen-Wei Xie, Di Chen, Feiwu Yu, Haiming Zhao, Jianxiao Yang, et al. Wan: Open and advanced large-scale video generative models. arXiv preprint arXiv:2503.20314, 2025.

---

> ### Author Response · Authors · 2025-11-27
>
> Dear Reviewer
>
> We hope this message finds you well. Since the discussion phase is nearing completion, we wanted to follow up and ensure all your concerns have been fully addressed. If there are any further points or clarifications you would like from us, please feel free to let us know—we would be grateful for any additional feedback.
>
> Thank you for your efforts in reviewing our work.

---

### Author Response · Authors · 2025-12-03

We sincerely thank the reviewers and chairs for their time and constructive feedback on our submission.

Our work focuses on a critical yet often overlooked problem in image editing: **physical realism**. Reviewers Ytia, ghns, and QMZg collectively found our research "highly meaningful," "critical," and "interesting and rarely explored." To systematically tackle this challenge, we introduce PICABench, a fine-grained benchmark covering eight physical sub-dimensions across Optics, Mechanics, and State Transitions, which reviewers recognized as "comprehensive" and "well-constructed." We further propose PICAEval, a novel and robust evaluation metric that leverages a region-grounded VLM-as-a-judge protocol to overcome the limitations of standard VLM-based scoring (reviewer guET found this "novel and effective"). Finally, we constructed PICA-100K, a large-scale synthetic dataset generated from video models, which reviewer Ytia highlighted as a "clever approach" to distilling physics into generative models. Furthermore, the clarity of our presentation was acknowledged, with reviewers guET and QMZg both finding the paper "well-organized and easy to follow."

Our rebuttal successfully addressed the reviewers’ main concerns regarding model generalization, distinctions from prior work, evaluation clarity, and frame-selection rationale. The initially negative reviewer (QMZg, 4) **raised their score to 6** after the rebuttal, explicitly stating that the majority of concerns had been addressed.

- **Generalization to other architectures.**

  Reviewers questioned whether the gains from PICA-100K were model-specific. We addressed this by fine-tuning Qwen-Image-Edit,  and observed consistent improvements across all physical dimensions and prompt levels. This confirms that PICA-100K transfers beyond a single architecture (reviewers Ytia, QMZg).

- **Generalization to other benchmarks**

  To further address concerns about overfitting, we evaluated the fine-tuned Qwen-Image-Edit on KRIS-Bench, where it achieved improved instruction-following performance. This demonstrates that PICA-100K also enhances models' broader editing capabilities (reviewer QMZg).

- **Clearer distinctions from prior works (Frame2Frame, ByteMorph).**

  We expanded Sec. 3.4 to explicitly compare motivations and pipelines: prior work focuses on non-rigid motion or inference-time editing, whereas our goal is *learning physics from videos*. We also highlight our fully synthetic, controlled pipeline and the improved background preservation it enables (reviewer Ytia).

- **Clarifications of definitions and metrics.**

  We added detailed explanations of hallucination, the role of prompt specificity, the Acc/Cons metrics, and the trade-off between physical realism and background preservation. These clarifications address concerns raised by reviewer guET.

- **Unified model analysis and frame-selection rationale.**

  We provide new qualitative results showing that unified models (e.g., Bagel) can reason correctly but still fail to generate physically plausible edits—supporting our claim that current models lack internalized physics. We also explain why using only the first/last frames avoids propagating instability from intermediate frames (reviewer ghns).

Sincerely,

The Authors of Submission 1871

---

### Meta-Review · Area_Chair_3UoU · 2026-01-07

**Summary:**

This paper introduces PICABench, a benchmark designed to evaluate physical realism in instruction-based image editing, addressing an important gap where existing benchmarks focus primarily on semantic fidelity while overlooking physical effects such as shadows, reflections, and object interactions. The work proposes a fine-grained taxonomy spanning three core dimensions (Optics, Mechanics, State Transition) with eight sub-dimensions, along with PICAEval, a region-grounded VLM-as-a-judge evaluation protocol that incorporates human-annotated key regions to reduce hallucination and improve assessment reliability. Additionally, the authors construct PICA-100K, a synthetic dataset derived from video generation models, and demonstrate that fine-tuning on this dataset improves physical consistency of baseline models.

All four reviewers acknowledged the importance and novelty of studying physical realism in image editing, praising the well-structured benchmark design, clear presentation, and comprehensive evaluation across 11 models. However, significant concerns were raised regarding: (1) the limited performance gains from the proposed PICA-100K dataset, (2) questions about the novelty of the video-based data construction pipeline relative to prior works (ByteMorph, Frame2Frame), (3) the choice of base model for fine-tuning experiments, (4) potential confusion between physical reasoning and long-text comprehension abilities, and (5) concerns about evaluation metric clarity and data generalization beyond PICABench.

---
This paper makes a timely and valuable contribution by systematically addressing the underexplored problem of physical realism in image editing. The core strengths that support acceptance include:

Novel and Important Research Direction: The paper identifies a critical gap in existing benchmarks and models, providing the first comprehensive framework for evaluating physics-aware image editing across multiple dimensions.

Well-Designed Benchmark: PICABench offers a fine-grained, systematic evaluation framework with concrete, checkable criteria across eight sub-dimensions, which will benefit the community in advancing beyond semantic-only editing.

Reliable Evaluation Protocol: PICAEval's region-grounded, QA-based approach represents a methodological improvement over generic VLM-as-judge setups, with demonstrated alignment to human assessments.

Thorough Experimental Analysis: The comprehensive evaluation of 11 open- and closed-source models provides valuable insights into the current state of physical realism in image editing.

Responsive Rebuttal: The authors addressed most major concerns with additional experiments (Qwen-Image-Edit fine-tuning, Kris-Bench evaluation, Uni-CoT comparison) and clear clarifications.

The remaining concerns—primarily the limited performance gains from PICA-100K and the unresolved questions about physical reasoning vs. instruction following—are acknowledged limitations rather than fatal flaws. The paper's primary contribution is diagnostic (revealing the gap) rather than prescriptive (solving the gap), which is appropriate for a benchmark paper. The observation that even state-of-the-art models struggle with physical realism is itself a valuable finding that motivates future research.

The reliance on GPT-5 for evaluation is a methodological concern that the authors should address in the camera-ready version by providing detailed logging of API calls, exploring open-source VLM alternatives for reproducibility, or committing to releasing evaluation snapshots.

Overall, the paper's contributions to establishing a new evaluation standard for physical realism in image editing outweigh its limitations, making it suitable for acceptance as a poster presentation.

**Reviewer Concerns:**

Concerns Adequately Addressed by Rebuttal:
Research Motivation and Contribution Positioning (Reviewer Ytia, QMZg): The authors successfully clarified that their core contribution is the diagnostic benchmark rather than a definitive solution, and that PICA-100K aims to validate whether models can learn physics from synthetic video data rather than surpassing SOTA models. This reframing appropriately sets expectations.

Generalization Across Base Models (Reviewer Ytia, QMZg): The authors provided additional experiments fine-tuning Qwen-Image-Edit on PICA-100K, demonstrating consistent improvements across both pure diffusion models (FLUX.1 Kontext) and VLM+Diffusion architectures. This effectively addresses concerns about dataset specificity to a single architecture.

Differentiation from Prior Works (Reviewer Ytia): The authors provided a clear comparison with ByteMorph and Frame2Frame, articulating distinct motivations (physical realism vs. non-rigid editing/zero-shot feasibility), methodological paradigms (offline data construction vs. training-free inference), and pipeline differences (fully synthetic with background consistency preservation). This adequately distinguishes their contribution.

Definition of Physical Hallucination (Reviewer guET): The authors clarified the criteria for judging physical violations, distinguishing between "physically possible airborne motion" and "unsupported hovering" with reference to their causality definition. This provides necessary conceptual clarity.

Evaluation Metric Clarification (Reviewer ghns): The authors corrected the misunderstanding regarding Chamfer Distance and F-Score, clarifying that PICAEval (Acc) and PSNR (Con) are the primary metrics, and committed to providing detailed metric explanations in the revised version.

Multi-step Reasoning Comparison (Reviewer guET): The authors provided Uni-CoT comparison results showing that multi-step reasoning alone is insufficient for physical realism, strengthening the argument that physics principles need to be internalized rather than reasoned at inference time.

Cross-benchmark Generalization (Reviewer QMZg): The authors tested their fine-tuned model on Kris-Bench, showing improvements in instruction following without overfitting to PICABench. This partially addresses generalization concerns.

Unified Model Analysis (Reviewer ghns): The authors provided qualitative analysis of Bagel in thinking mode, showing that it can reason correctly but fails to generate physically consistent outputs, strengthening the claim about decoupled world knowledge and generation capabilities.

Concerns Not Fully Resolved:
Limited Performance Gains (All Reviewers): While the authors reframed PICA-100K as hypothesis validation rather than a solution, the fundamental concern remains that the proposed approach yields marginal improvements (+2-3% accuracy). The fine-tuned model still underperforms top-tier closed-source models (GPT-Image-1, Seedream 4.0), limiting the practical impact of this contribution.

Physical Reasoning vs. Long-Text Comprehension (Reviewer guET, ghns): The authors did not directly address whether the performance differences across prompt levels reflect genuine physical reasoning capabilities or simply better long-text instruction following. The observation that explicit prompts yield significantly better results than intermediate prompts could indicate models are instruction followers rather than physics reasoners, but the authors agreed with this interpretation rather than proposing how to disentangle these factors.

State Transition Performance Degradation (Reviewer ghns): The authors acknowledged that using only first/last frames may fail to capture state transition coherence, but provided no concrete solution beyond stating they will "continue exploring better strategies." This remains a methodological weakness in the data pipeline.

Accuracy-Consistency Trade-off (Reviewer guET): The authors explained that more detailed prompts encourage broader scene modifications, naturally reducing consistency. However, this represents an unresolved limitation—an ideal physics-aware model should improve physical accuracy while maintaining stability in non-edited regions.

Reliance on Closed-Source VLM for Evaluation (Implicit Concern): The use of GPT-5 as the evaluation backbone raises significant concerns about reproducibility, accessibility, and long-term stability of the benchmark. The API may be updated silently, results may not be reproducible across time, and access barriers exist for researchers in certain regions or with limited resources. This is a fundamental methodological concern for a benchmark paper that was not explicitly raised by reviewers but represents a significant limitation.

**Reviewer Scores:**

Reviewer Ytia (Original: 6 - marginally above acceptance):
The author rebuttal adequately addressed concerns about fine-tuning other base models and comparison with prior works. However, the limited performance gains remain a concern. The reviewer would likely maintain their score at 6, as the core limitation (small gains) was reframed rather than resolved.

Reviewer guET (Original: 6 - marginally above acceptance):
The clarifications on hallucination definition, prompt-level motivation, and metric explanations were helpful. The Uni-CoT comparison strengthened the paper. However, concerns about physical reasoning vs. long-text comprehension were not fully disentangled. The reviewer would likely maintain their score at 6.

Reviewer ghns (Original: 4 - marginally below acceptance):
The qualitative analysis of unified models and clarification on metrics addressed some concerns. However, the first/last-frame limitation remains unresolved, and the paper still lacks a path toward models that can generalize physical principles from scene context. The reviewer might slightly increase their score to 5, acknowledging the additional experiments but remaining concerned about fundamental limitations.

Reviewer QMZg (Original: 4 - marginally below acceptance):
The cross-benchmark evaluation on Kris-Bench and additional Qwen-Image-Edit experiments partially address generalization and model choice concerns. However, the reviewer's fundamental concern about whether pure diffusion models are the right paradigm for physics-aware editing was not fully resolved. The reviewer might increase their score to 5, but would likely remain below the acceptance threshold. The reviewer claim the authors have addressed the proposed questions, and will improve his scores. So I think this final score of the reviewer QMZg will be 5 points or higher.

---

### Decision · Program_Chairs · 2026-01-26

Accept (Poster)